# Offline evaluation in RL: soft stability weighting to combine fitted Q-learning and model-based methods

## Abstract

The goal of offline policy evaluation (OPE) is to evaluate target policies based on logged data under a different distribution. Because no one method is uniformly best, model selection is important, but difficult without online exploration. We propose *soft stability weighting (SSW)* for adaptively combining offline estimates from ensembles of fitted-Q-evaluation (FQE) and model-based evaluation methods generated by different random initializations of neural networks. Soft stability weighting computes a state-action-conditional weighted average of the median FQE and model-based prediction by normalizing the state-action-conditional standard deviation of ensembles of both methods relative to the average standard deviation of each method. Therefore it compares the relative stability of predictions in the ensemble to the perturbations from random initializations, drawn from a truncated normal distribution scaled by the input feature size. We extend this approach to soft stability weighting via partial rollouts (SSWPR), which introduces weights over different timesteps corresponding to partial rollouts. We show on two simulated environments that both FQE and model-based approaches have systematic errors in different regions of the state space and our soft stability weighting metric provides a signal as to which method achieves less state/action-conditional error, suggesting benefits from our approach. Soft-stability weighting outperforms simple averaging of fitted-Q-evaluation and model-based estimates, improves upon both approaches half of the time, and is never the worst. Although our experiments focus on FQE and model-based approaches, SSW can be used to combine other and more methods.

## Soft Stability Weighting

**Introduction** In many real-world applications of reinforcement learning (RL), policy optimization through online interaction with the environment is impractical or impossible due to constraints on safety, performance, or time. In such settings, one key challenge is *off-policy evaluation* (OPE): estimating the value of a target policy based on batch data without the ability to collect data online. A unique challenge is that it is not possible to validate such estimates, since it requires running policies on the real environment [14]. Many algorithms have been proposed for offline evaluation of reinforcement learning agents, including importance sampling methods, doubly robust methods, fitted q-evaluation (FQE), and model-based evaluation (MBE) [11, 13, 9, 3, 12]. (See Appendix B for a more in-depth discussion of related work). But, performance across different evaluation methods can vary greatly across different types of reinforcement learning tasks. As a consequence, model selection is necessary but difficult compared to supervised learning because typical out-of-sample validation is not possible.

**Problem Setup** Please see Appendix A for a complete description. The environment is a Markov Decision Process $\mathcal{M} = (\mathcal{S}, \mathcal{A}, p, r, \mu_0, \gamma)$ with state-space $\mathcal{S}$, action space $\mathcal{A}$, $p(s'|s, a)$ is the

transition function given state $s$ and action $a$, $r(s, a)$ is reward given a state $s$ and action $a$, $\mu_0(s)$. Given a policy $\pi(a|s)$, the value of $\pi$ is defined to be $v(\pi) = \mathbb{E}_{s \sim \mu_0}[V^\pi(s)]$ where $V^\pi(s) = \mathbb{E}\left[\sum_{t=1}^{\infty} \gamma^{t-1} r_t | s\right]$ is the state value function in the infinite horizon setting. The Q-function is $Q^\pi(s, a) = \mathbb{E}_{s' \sim p(s,a)}(r(s, a) + \gamma V(s'))$. Fitted-Q-evaluation iteratively fits a Q function from the logged data: $Q^\pi(s, a) = \mathbb{E}^\pi_{s' \sim p(s,a)}(\sum_{i=1}^{N} \gamma^{i-1} r_i | s_1 = s, a_1 = a)$,. The Q function parametrized by $\theta$ fits Bellman residuals from the previous iteration's parameter $\theta'$. Model based evaluation (MBE) fits neural networks for the rewards and transition dynamics models.

**Soft Stability Weighting (SSW)** We propose two methods for adaptively weighting the estimates from FQE and MBE: Soft Stability Weighting (SSW) and Soft Stability Weighting via Partial Rollouts (SSWPR). Even though this method can be used to combine any number of evaluation methods, we focus on FQE and model-based evaluation (MBE), since both are popularly used [5]. Our metric is based on ensembles for each individual method, obtained by random initializations of the neural networks used in each method.[1] These ensembles are informative of the stability of predictions to arbitrary random initializations, drawn from a truncated normal distribution standardized by feature size. We measure stability of the state-action $(s, a)$ conditional predictions by normalizing the standard deviation of the $(s, a)$-conditional predictions of a method (over the predictions of the ensemble from random initializations) relative to the quantile (over the population of $(s, a)$ tuples) of the $(s, a)$-conditional standard deviation. We evaluate the $(s, a)$-conditional standard deviation over the predictions of the ensemble, i.e. the random initializations. We denote the $(s, a)$-conditional standard deviation of ensemble predictions for a method $m \in \{\text{FQE}, \text{MB}\}$ as $\hat{\sigma}_m(s, a)$. In order to compare the standard deviation of $(s, a)$-conditional predictions and assess which method is less stable, we also normalize the standard deviation by subtracting the lower marginal $c-$quantile of the conditional standard deviation (marginalizing over the distribution of $(s, a)$) and dividing by the interquantile range. The overall stability metric for FQE is

$$u_{\text{FQE}}(s, a) = \min\left(1, \max\left(\frac{\hat{\sigma}_{\text{FQE}}(s, a) - \text{quantile}_{\mathbf{D}_{\pi_b}}(\hat{\sigma}_{\text{FQE}}, c)}{\text{quantile}_{\mathbf{D}_{\pi_b}}(\hat{\sigma}_{\text{FQE}}, 1 - c) - \text{quantile}_{\mathbf{D}_{\pi_b}}(\hat{\sigma}_{\text{FQE}}, c)}, 0\right)\right),$$

where $c < 1/2$ because we normalize by an interquantile range. We additionally truncate the stability metric by 0 and 1. Once both $u_{\text{FQE}}(s, a)$ and $u_{\text{MB}}(s, a)$ are computed, we can compute the weight $\alpha$ as $\alpha(s, a) = \sigma(\log(\frac{u_{\text{FQE}}(s,a)}{u_{\text{MB}}(s,a)}))$, where $\sigma$ is the sigmoid function and $0 \leq \alpha(s, a) \leq 1$. The final weighting that comprises SSW, derived from the conditional standard deviation of predictions over the ensemble of random initializations, is

$$\text{SSW}(s, a) = (1 - \alpha(s, a))Q_{\text{FQE}}(s, a) + \alpha(s, a)Q_{\text{MB}}(s, a).$$

Intuitively, $\alpha(s, a)$ biases towards 0 and puts more weight on $Q_{FQE}(s, a)$ if the stability metric of the FQE ensemble is low relative to that of the model-based method. $\alpha(s, a)$ biases towards 1 and puts more weight on $Q_{\text{MB}}(s, a)$ if the stability metric of the FQE ensemble is high relative to that of the model-based method. In the case that the stability metric for each method is similar, $\alpha(s, a)$ puts near equal weight on $Q_{\text{FQE}}(s, a)$ and $Q_{\text{MB}}(s, a)$. Lastly, we weight between median values of each ensemble's predictions due to the robustness of the median to outliers. SSW generalizes to different and additional methods by replacing the $\alpha(s, a)$ normalization to $[0, 1]$ with a softmax operator.

**Soft stability weighting via partial rollouts (SSWPR)** We propose a second weighting method based on *partial rollouts* - rollouts from a dynamics model that are terminated at step $k$ before the end of a trajectory. Given an ensemble of dynamics models, an ensemble of FQE models, and a specific state-action pair $(s, a)$ that we are querying, we compute independent partial rollouts up to $k$ steps starting from $(s, a)$ from each of the dynamics models in the given ensemble. At each simulated step of each rollout, we compute the $\text{SSW}(s_t^i, a_t^i)$ between individual pairs of FQE and dynamics models where $(s_t^i, a_t^i)$ is the state and action at time $t$ in the $i$th partial rollout as well as $\text{CR}_t^i$ which is the discounted sum of rewards of partial rollout $i$ up to time $t$. Given a time point $t$ and a partial rollout $i$, we then have an estimate of the value of $(s, a)$ given as $\hat{v}_t^i(s, a) =$

---

[1] For FQE, we fit (iteratively) the Q-function; we initiate randomly a multi-layer perception with 2 residual blocks and a hidden layer size of 50. For model-based approaches, we randomly initialize two neural networks with hidden layer size of 200 and 3 hidden layers to estimate the transition dynamics and reward function. See the appendix for more details.

CR$_t^i + \gamma^t * $SSW$(s_t^i, a_t^i)$. We compute the standard deviation of predictions at $(s, a)$ across all partial rollouts for a fixed $t$ as $u_t = std(\hat{v}_t(s, a))$. We can then compute weights using the softmax function across $t$: $\alpha_t = \text{softmax}(-u)_t$, where $\sum_{t=1}^k \alpha_t = 1$. We compute final value estimate for $(s, a)$ as $\sum_{t=1}^k \alpha_t * \text{median}(\hat{v}_t(s, a))$, where the median is computed across partial rollouts. Intuitively, we can interpret the SSWPR estimate as weighting across time where estimates at each time point interpolate between the SSW estimate at time $t$ in the partial rollout and the collected rewards from partial rollout $i$ up to time $t$.

Unlike SSW, SSWPR can leverage value estimates from simulated states local to the original state that is being conditioned upon. If the value estimates at a simulated state at time $t$ are more stable, then more weight is placed on $\text{median}(\hat{v}_t(s, a))$ relative to other times and vice versa. The full algorithm is given below.[2]

**Algorithm:** Inputs: Ensemble size of $N_{\text{ens}}$, ensemble of FQE models $\{\hat{Q}_i^{\text{FQE}}\}_{1:N_{\text{ens}}}$, ensemble of dynamics models $\{(\hat{P}_i, \hat{r}_i)\}_{1:N_{\text{ens}}}$ where $\hat{P}$ denotes the transition model and $\hat{r}$ denotes the rewards model, a maximum horizon of $k$, and a given state and action $(s, a)$

1. For $i$ in 1 to $N_{\text{ens}}$ :

    (a) Compute partial rollout PR$_i$ from transition model $\hat{P}_i$ for up to $k$ simulated steps

    (b) For $j$ in 1 to $N_{\text{ens}}$ :

        i. Compute SSW$(s_t^i, a_t^i)$ using FQE model $\hat{Q}_j^{\text{FQE}}$ and dynamics model $(\hat{P}_j, \hat{r}_j)$ for each time step $t$ in $PR_i$

        ii. Compute $(s, a)$-conditional value, $\hat{v}_t^i(s, a) = $CR$_t^i + \gamma^t$ SSW$(s_t^i, a_t^i)$ where CR$_t^i = \sum_{j=1}^t \gamma^{t-1}\hat{r}_i$ are the collected rewards up to time $t$

2. Compute standard deviation and median $\hat{\sigma}(\hat{v}_t(s, a)), \text{median}(\hat{v}_t(s, a))$ across rollouts for $t$ in 1 to $k$

3. Compute weights $\alpha_t = \text{softmax}(-\hat{\sigma}(\hat{v}_t(s, a)))_t$

4. Output $\sum_{t=1}^k \alpha_t \cdot \text{median}(\hat{v}_t(s, a))$ as the final conditional value estimate

# Experimental Results

We give a brief description here of our experimental setup in two simulated RL environment tasks: 2D World and Mountain Car[3]. More details on the experimental setup are in the Appendix. Baseline methods include simple ensembles of either approach, and simple averaging. These are reasonable baselines because using either approach is quite common; and comparing against simple averaging shows whether the stability weighting mechanism has any performance boosts compared to naive, non-adaptive combination. We compare against an ensemble of FQE models and an ensemble of dynamics models for model-based evaluation. We take a simple average of conditional estimates from the FQE ensemble and the ensemble of dynamics models as a further baseline. The last (un-achievable) "skyline" is an adaptive oracle selection between FQE and model-based method where value estimates conditioned on the state come from the method with the lower error. This final comparison is unachievable in practice but is a useful benchmark for the potential improvement of the two proposed methods.

To evaluate each method trained using the dataset of a particular behavior policy and a given evaluation policy, we first sample $500$ $(s, a, r, s')$ tuples from Monte Carlo rollouts using the evaluation policy on the oracle environment. This setup allows us to evaluate how accurately the offline evaluation methods can estimate the value of the evaluation policy throughout the state-space where the evaluation policy is likely to visit. We compute conditional value estimates using each of the methods and compare them against the estimate using Monte Carlo rollouts using the target evaluation policy on the oracle environment. The latter estimate serves as ground-truth to compare the offline evaluation methods against. This is done for each pair of behavior and evaluation policies.

---

[2]Note that for the experiments in this work, we used $k = 5$ steps for the partial rollout length. We chose to use a smaller value of 5 compared to the max time step of 300 in the 2D-world environment due to the computational cost of querying value estimates on partial rollouts. However, in our stability experiments we compare against using values of $k = 3$ and $k = 7$ to check the robustness of SSWPR to a perturbation of this parameter.

[3]The 2D world is a two-dimensional continuous state- and action-space environment designed with piece-wise heterogeneity in underlying models and variance. The mountain car environment is a standard one-dimensional state space environment used in evaluation of offline methods.

Table 1: Mean absolute error for conditional value estimates from $(s, a, r, s')$ tuples from Monte Carlo rollouts of the evaluation policy on the 2DWorld environment)

| $\pi_e$ | $\pi_b$ | FQE | MB | Avg. FQE,MB | Oracle | SSW | SSWPR |
|---|---|---|---|---|---|---|---|
| $\pi_{b_1}$ | $\pi_{b_1}$ | 5.9 | 3.7 | 4.0 | 3.7 | 4.4 | 4.1 |
| $\pi_{b_1}$ | $\pi_{e_1}$ | 5.0 | 3.8 | 3.2 | 3.8 | 3.5 | 3.9 |
| $\pi_{b_1}$ | $\pi_{e_2}$ | 13.3 | 8.7 | 9.3 | 8.7 | 8.6 | 8.6 |
| $\pi_{b_1}$ | $\pi_{e_3}$ | 18.0 | 51.1 | 21.4 | 18.0 | 11.8 | 11.9 |
| $\pi_{b_2}$ | $\pi_{b_2}$ | 10.5 | 5.4 | 7.4 | 5.1 | 6.3 | 6.0 |
| $\pi_{b_2}$ | $\pi_{e_1}$ | 20.1 | 8.1 | 14.1 | 8.1 | 8.6 | 8.6 |
| $\pi_{b_2}$ | $\pi_{e_2}$ | 11.5 | 8.1 | 14.1 | 8.1 | 13.5 | 14.6 |
| $\pi_{b_2}$ | $\pi_{e_3}$ | 17.1 | 31.1 | 17.3 | 17.1 | 17.8 | 16.1 |

Table 2: Error of conditional value estimates vs. Monte Carlo rollouts of the evaluation policy on the 2DWorld environment

| Method | MAE | # best | # worst | # outp |
|---|---|---|---|---|
| FQE | 12.7 | 0 | 6 | NA |
| MB | 15.1 | 3 | 2 | NA |
| FQE+MB | 10.6 | 1 | 0 | 2 |
| SSW | 9.2 | 3 | 0 | 4 |
| SSWPR | 9.2 | 2 | 0 | 3 |

Table 3: Error of conditional value estimates vs. Monte Carlo rollouts of the evaluation policy on the Mountain Car environment

| Method | MAE | # best | # worst | # outp. |
|---|---|---|---|---|
| FQE | 10.7 | 0 | 7 | NA |
| MB | 7.1 | 2 | 1 | NA |
| FQE+MB | 8.2 | 1 | 0 | 2 |
| SSW | 6.5 | 3 | 0 | 6 |
| SSWPR | 6.3 | 5 | 0 | 5 |

The results for each pair of behavior ($\pi_b$) and evaluation ($\pi_e$) policies are shown in Table 1 for the 2D World task, and summary results across pairs are shown in Tables 2 and 3 for 2D World and Mountain Car tasks. We include the performance of FQE and MB ensembles, the simple average (FQE + MB), the oracle skyline (Oracle), and our methods SSW and SSWPR. The columns indicate, over a wide range of pairs of evaluation and behavior policies, the number of times each method has the best MAE, worst; and for SSW, SSWPR, how many times they *outperform both* FQE and model-based methods. Empirically, SSW and SSWPR have the lowest mean average error of 9.2 and 9.1, respectively, across pairs of behavior and evaluation policies. Note that SSW and SSWPR outperform both FQE and the model-based method individually, which achieves average errors of 12.7 and 15.0. Our results also show that neither SSW and SSWPR are outperformed by both FQE and the model-based method on any pair of behavior and evaluation policy. Half the time, SSW and SSWPR outperform both FQE and the model-based method. Lastly, SSW and SSWPR outperform a simple averaging of FQE and MBE, which achieves an average error of 10.6.

We also show additional results in the appendix that visualize individual predictions of SSW, SSWPR, FQE, and the model-based method against the target values in figs. 6 to 13. Qualitatively, we see that in cases where the model-based method and FQE are biased in opposite directions, SSW and SSWPR tend to outperform both FQE and the model-based method, exemplified by Figures 10 and 12. We also see that SSW and SSWPR tend to reduce the extremity of outlier values produced by either the model-based method or FQE, shown by Figures 9, 10 and 12. In the case that FQE and the model-based method are biased in the same direction, SSW and SSWPR tend to have less utility as shown in Figure 11.We additionally include histogram of absolute errors of SSW, SSWPR, FQE, and the model-based method in Figures 14 and 16 to 21 and appendix D. Visually, we see that SSW and SSW tend to produce less extreme absolute errors compared with FQE and the model-based method as shown in Figures 17, 19 and 21. Both stability methods give lower errors in many cases unlike FQE as shown in Figures 17, 19 and 21. We note that SSW and SSWPR have similar errors on average.

**Conclusion** In conclusion, we have proposed SSW and SSWPR to combine methods based on state-action-conditional standard deviation of their predictions (normalized by average variability). We show that the stability metric is informative, and in experiments on two simulated environments, our state-adaptive weighting often outperforms both FQE and model-based methods, and is never the worst. The same adaptive weighting scheme can of course be adapted to other types of methods, although we have investigated the two most popular approaches, as well as additional methods.

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

## A  Problem Setup

The environment is a Markov Decision Process $\mathcal{M} = (\mathcal{S}, \mathcal{A}, p, r, \mu_0, \gamma)$ where $\mathcal{S}$ is the state-space, $\mathcal{A}$ is the action space, $p(s'|s, a)$ is the transition function given state $s$ and action $a$, $r(s, a)$ denotes the reward function given a state $s$ and action $a$, $\mu_0(s)$ is the initial state distribution, and $\gamma \in [0, 1]$ is the discount factor. At each time step $t$, the agent receives some state $s_t$ and selects an action $a_t$ via $\pi(a_t|s_t)$ to take and receives reward $r_t$ and the next state $s_{t+1}$ from the environment.

Given a policy $\pi(a|s)$, the value of $\pi$ is defined to be $v(\pi) = \mathbb{E}_{s \sim \mu_0}[V^\pi(s)]$ where

$$V^\pi(s) = \mathbb{E}_{s' \sim p(s,a), a \sim \pi}\left[\sum_{t=1}^\infty \gamma^{t-1} r_t | s\right]$$

is the state value function in the infinite horizon setting. Related is the Q-function which restricts the action taken at state $s$ and is defined to be

$$Q^\pi(s, a) = \mathbb{E}_{s' \sim p(s,a)}(r(s, a) + \gamma V(s')).$$

In the typical reinforcement learning setting, the goal is to train a policy $\pi$ such that the value function is maximized. An additional goal in reinforcement learning is to evaluate the value of a policy $\pi$. The aim is to compute an estimate of $V^\pi(s)$ where $s$ is directly given or where $s \sim \mu_0$. We focus on the offline reinforcement learning setting where one only has access to logged data of the form $\mathcal{D} = (s_i, a_i, r_i, s'_i)$ and access to the real environment $\mathcal{M}$ is not available [14]. The logged data is derived from one or more behavior policies which is denoted by $\pi_b$. In the episodic reinforcement learning scenario, the logged data can also be written as $\mathcal{D} = \{\tau_i\}$ where $\tau = (s_1, a_1, r_1, s'_1, \ldots, s_n, a_n, r_n, s'_N)$ where the length of the episode $N$ can vary across episodes. In the typical setting, the logged data is static and assumed to come from the target environment. Additional data collection from the actual environment is not possible.

The goal of offline evaluation or off-policy evaluation is to estimate $v(\pi_e)$ where $\pi_e$ denotes the evaluation policy. The evaluation policy is the policy which we want to estimate the value using the logged data $\mathcal{D}$ deriving from the behavior policy. Offline evaluation is especially difficult because the goal is estimate the value of a policy which is typically not represented in the logged data.

## B  Related work

**Fitted Q-evaluation**

Fitted Q-evaluation (FQE) is a off-policy temporal difference learning algorithm based on a slight variation of the fitted Q-iteration algorithm [11]. FQE involves learning the following Q function from the logged data:

$$Q^\pi(s, a) = \mathbb{E}^\pi_{s' \sim p(s,a)}\left(\sum_{i=1}^N \gamma^{i-1} r_i | s_1 = s, a_1 = a\right),$$

which can intuitively be interpreted as the value of taking action $a$ at state $s$ and then following policy $\pi$ for the rest of the trajectory. Note that the value of the policy can be written in terms of the Q-function $\mathbb{E}[Q^\pi(s, \pi(s))]$ where $s \sim \mu_0$. The Q-function is trained by minimizing the following

$$\mathbb{E}_{(s,a,r,s') \sim \mathbf{D}}[(Q_\theta(s, a) - r - \gamma Q_{\theta'}(s', \pi(s')))^2],$$

where $\theta$ are the parameters of the function class used to approximate the Q-function and $\theta'$ are the parameter values in the previous iteration of the training process.

FQE is typically implemented as shown below [11]. We assume an evaluation policy $\pi_e$, a function class $\mathbb{F}$, and a dataset $D = \{(s_i, a_i, r_i, s'_i)\}_{i=1}^n$. The algorithm proceeds as follows:

    1. Randomly initialize parameters of $Q_0^{\pi_e} \in \mathbb{F}$

    2. for $k$ from 1 to $K$

    3.   (a) Compute FQE target $y_i = r_i + \gamma Q_{k-1}^{\pi_e}(s'_i, \pi_e(s'_i))$ for every $i$

        (b) Construct training data as follows: $D_{FQE_k} = \{(s_i, a_i, y_i)\}_{i=1}^n$

        (c) Solve $Q_k^{\pi_e} = argmin_{f \in \mathbb{F}} \frac{1}{n} \sum_{i=1}^n (f(s_i, a_i) - y_i)^2$

    4. Output $Q_K^{\pi_e}$

**Model-based evaluation**

Model-based (MB) evaluation is similar to model-based reinforcement learning in that it involves learning a simulation of the real environment $\mathcal{M}$. More specifically, both the transitions $p(s'|s, a)$ and the reward function $r(s, a)$ are learned via the logged data $\mathcal{D}$ using standard supervised learning techniques. The fitted transition and reward functions are then used to simulate trajectories using the behavior policy. The observed rewards of the simulated trajectories can then be used to calculate values for the behavior policy. These trajectories can be referred to as monte carlo rollouts.

To estimate the Q-value using the fitted reward and transition models, we have that, where $s' \sim \hat{p}(s, a)$,

$$\hat{Q}^\pi_{MB}(s, a) = \sum_{t=0}^N \hat{r}(s, a) + \gamma \hat{V}_{MB}(s')), \qquad V^\pi_{MB}(s) = \mathbb{E}_{s' \sim \hat{p}(s,a), a \sim \pi_e} \left[ \sum_{t=1}^N \gamma^{t-1} \hat{r}_t | s \right].$$

**Related Work**

Many algorithms have been proposed to do offline evaluation of reinforcement learning agents, including importance sampling methods, doubly robust methods, fitted q-evaluation (FQE), and model-based evaluation [11, 13, 9, 3, 12]. FQE has been studied theoretically in a variety of literature [4, 6], including its dependence on theoretical assumptions of concentratability (coverage/sequential overlap) and Bellman completeness. In practice, FQE has found more empirical success compared to importance sampling and doubly robust methods due to the lower variance of the estimate and generalizability from function approximation. Recently, Zhang et al. [16] focused on FQE with general and differential function approximators using Z-estimation theory. Among other analysis, they show the FQE estimation error is asymptotically normal, justifying the bootstrap.

**FQE vs. model-based methods**   Computing value estimates using FQE does not rely on simulating entire rollouts unlike the model-based method, where errors can compound if the horizon is especially long. However, a downside to FQE is that is unclear how to tune the parameters and architecture of FQE if function approximation is used. The model-based method performs well when the environment transition and reward functions are simple and can be easily approximated through function approximation. It is typically easier to tune the hyperparameters of dynamics models, compared to tuning FQE models, since a validation loss based on the observed transitions and rewards can be computed on a hold-out set. A downside to the model-based method is that estimates derived from simulated trajectories may compound errors over time if the maximum horizon length is long. This is the case since values are calculated from rewards along trajectories which are simulated autoregressively for the model-based method. This is not the case for FQE, which directly output q-values.

**Model selection in offline RL**   A line of recent work investigates approaches for model selection with varying degrees of algorithmic complexity [17]. We empirically investigate weighting-based approaches that are algorithmically simple, based almost entirely on oracle function evaluation access to ensembles of candidate models. Our idea of combining estimators is inspired from previous work in weighted online learning, where past predictability is used to weight different online predictors [2, 1]. Cesa-Bianchi et al. [2] weight online boolean predictors using exponential weighting computing using the number of mistakes made by each predictor in the past. Altieri et al. [1] propose the CLEP (combined linear and expoential predictors) algorithm for forecasting covid-19 cases and deaths. CLEP weights each predictor based on recent predictive performance, where more accurate predictors are assigned higher weights. Unlike these works, we use the stability of ensembles conditioned on the state-space as a weighting mechanism instead of local or past predictive performance.

**PCS (Predictability, computatibility, and stability)**

The PCS (Predictability, computatibility, and stability) framework [15] outlines principles for a data science problem and an approach with an underlying aim of providing reliable, responsible, and transparent results in the data science life cycle. Many of the ideas outlined in the PCS framework are very applicable to this data-driven setting of reinforcement learning. For example, predictability is an important reality check when working with logged data. Before extrapolating results to the

actual environment, reality checks on the logged data and trained policies or critics are required especially in high-stakes data problems that motivate offline reinforcement learning, such as clinical decision making, autonomous driving, and robotics.

Stability is also an essential check as results should be reproducible to small perturbations to data and models. Stability is especially important in reinforcement learning where the performance of particular algorithms rely on careful tuning and tricks in practice. Our work is tied to the stability principle in PCS as the main notion underlying the methodology is inspired by the stability of models. We further apply this principle to test the robustness of our proposed methods to human judgement calls in our experiments which can potentially impact results and conclusions.

## Experimental details: 2D Gridworld environment

In this section, we detail the simulator we created to benchmark the offline evaluation, the behavior policies, the evaluation policies, and the logged data generation process.

### Environment

The environment is depicted in Figure 1. The agent observes its position relative to the $x$ and $y$ axes, its horizontal and vertical velocities, and the time step. Each episode has a maximum horizon of 300 time steps and terminates when the time step reaches the maximum horizon or when the agent reaches the goal ( upper right corner given by $x \geq 4$ and $y \geq 4$). Note that the agent's $x$ and $y$ positions as well as the velocities are continuous values. The boundaries of the environment are given by the following lines: $x = 0, y = 0, x = 5, y = 5$. The agent is within the boundaries at all times. At each time step, the agent receives a negative reward conditioned on its $x$ and $y$ position outlined in Figure 5.3. If the agent successfully reaches the goal, the agent receives a completion reward of $+10$. At each step, the agent can choose from a set of 9 actions which correspond to moving to the left, right, up, down, and neutral as well as combinations of the horizontal and vertical moves.

The transition dynamics of the environment is detailed as follows. The horizontal and vertical velocities at time $t$ are outlined by

$$vel_t = \max(\min(vel_{t-1} + a_t * f, 0.1), -0.1),$$

where $f = 0.001, a_t \in \{-1, 0, 1\}$. The velocities then affect the horizontal and vertical positioning as follows:

$$pos_t = \max(\min(pos_{t-1} + vel_t, 5), 0).$$

If the agent hits the horizontal or vertical boundaries of the environment, its corresponding directional velocity is set to 0. Note that the transitions and rewards are deterministic. However, the starting state of the agent is stochastic. The agent's $x$ and $y$ positions are uniformly sampled from $[0, 5/6]$ at time step $t = 1$, while the horizontal and vertical velocities are set to 0.

### Policies

Two behavior policies were constructed by training deep Q-networks on the 2D world environment in typical online fashion. A multi-layer perceptron (MLP) network with 2 hidden layers with a hidden layer size of 50 were trained using the ADAM optimizer [8] with a learning rate of 0.001 and a batch size of 32 for both behavior policies with varying number of updates and initializations. Stochasticity was artificially embedded into the resulting Q-networks by adding a random probability of 0.25 where the agent performs a random action instead of the action given by its Q-network. The resulting behavior policies $\pi_{b1}$ and $\pi_{b2}$ had online value estimates of $-79.4$ and $-83.4$, respectively. The vertical and horizontal positions over time of each behavior policy are given by Figure 5.4.

Three evaluation policies were constructed in a similar fashion as the behavior policies with differing number of updates and initializations. Unlike the behavior policies, stochasticity was not embedded into the policy. The resulting evaluation policies $\pi_{e1}$, $\pi_{e2}$, and $\pi_{e3}$ had online value estimates of $-72.5$, $-85.0$, and $-92.0$, respectively. The vertical and horizontal positions over time of each evaluation policy are given by Figure 5.5.

**Offline datasets**

Behavior policies $\pi_{b1}$ and $\pi_{b2}$ were used to generate two offline datasets from interacting with the actual environment. Datasets $D_{b1}$ and $D_{b2}$ were derived from running the coresppnding policy for 1000 episodes in total. Summary statistics, such as the size of the datasets and proportions of each action taken, are shown in Table 5.1.

# Experimental details: offline evaluation training

We pair up each evaluation policy with each behavior policy to produce 6 pairs of behavior and evaluation policies. We also include two pairs that consist of each behavior policy paired up with itself as the evaluation policy. The goal is to estimate the value of the evaluation policy using the logged data deriving from the corresponding behavior policy for each pair of behavior policy and evaluation policy. The two offline evaluation methods we consider are FQE and model-based evaluation outlined in the previous section.

**Fitted Q-evaluation**

We detail how we train a FQE model on the logged data $\mathbb{D}$. A MLP with 2 residual blocks and a hidden layer size of 50 was initialized randomly. The neural network is trained on the tuples of $\mathbb{D}$ using the ADAM optimizer [8] with a learning rate of $0.001$ using a batch size of 32. The FQE model was trained for a max number of iterations of 75000. At every 500 iterations, the temporal-difference error was computed on a validation set of 20 episodes from the logged data.

Typically in the supervised learning case, model selection is used via computing the target metric on a validation set. In this setting, we cannot compute the target metric on the validation set, because the validation set contains trajectories by following actions derived from the behavior policy. Instead, we can use the temporal difference error as a proxy for the target metric where the temporal difference error is defined by

$$Q_{\pi_e}(s, a) - r(s, a) - \gamma * Q_{\pi_e}(s', a'),$$

where $(s, a, r, s', a')$ denotes the state, action, reward, next state, and next action. Note that only the true $Q$ function of $\pi_e$ minimizes the temporal difference error. Early stopping was applied by computing the absolute difference of the mean of the last 5 temporal difference error estimates and the previous 5 starting at the previous index. Mathematically we denote the stopping condition as

$$|TD_{i-5,i} - TD_{i-6,i-1}| < 0.001.$$

**Model-based evaluation**

We now detail how we train a model which learns the dynamics of the environment (reward and transition functions) to estimate values of offline agents. We first split the logged data $\mathbb{D}$ into a training set $\mathbb{D}_{train}$ and a validation set $\mathbb{D}_{val}$. We then initialize two neural network models, one that learns the reward function $r(s, a)$ and another that learns the transition function $p(s, a)$. Both neural network models were initialized with a hidden layer size of 200, with 3 hidden layers, and a learning rate of $5e^{-4}$ using the ADAM optimizer [8].

Both the reward and transition models are trained in a typical supervised learning fashion unlike FQE with the data being organized as $\{((s, a), r)\}$ and $\{((s, a), s')\}$ for the rewards and dynamics model, respectively. The mean squared error was used as the loss function train both the rewards and transition model. Each model trained using a max number of iterations of 50000. Every 1000 steps the loss was calculated on the validation data and early stopping was applied with a patience of 7.

 **TABLES**

Table 4: Summary statistics for each offline dataset corresponding to behavior policies $\pi_{b1}$ and $\pi_{b2}$ on the 2DWorld environment

| Statistic | Dataset $D_{b1}$ | Dataset $D_{b2}$ |
|---|---|---|
| Dataset size | 160,745 | 156,253 |
| Mean steps per episode | 159.7 | 155.3 |
| Mean reward per episode | -159.9 | -177.7 |
| Proportion of time collecting -1 rewards | 0.97 | 0.83 |
| Proportion of time collecting -2 rewards | 0.02 | 0.14 |
| Proportion of time collecting -4 rewards | 0.0 | 0.02 |

Table 5: Errors for FQE and the model-based (MB) method on partition A (sampled tuples from the evaluation policy where FQE outperforms the MB method) and partition B (sampled tuples from the evaluation policy where the MB method outperforms FQE) using models trained on data corresponding to the behavior policy $\pi_{b_1}$) and the 2DWorld environment

| Evaluation policy | FQE error on partition A | Model-based error on partition A | FQE error on partition B | Model-based error on partition B |
|---|---|---|---|---|
| $\pi_{b_1}$ | 3.7 | 6.5 | 9.3 | 3.4 |
| $\pi_{e_1}$ | NA | NA | 11.6 | 2.8 |
| $\pi_{e_2}$ | 8.0 | 13.4 | 16.3 | 4.9 |
| $\pi_{e_3}$ | 19.1 | 83.5 | 18.0 | 5.3 |

Table 6: Errors for FQE and the model-based (MB) method on partition A (sampled tuples from the evaluation policy where FQE outperforms the MB method) and partition B (sampled tuples from the evaluation policy where the MB method outperforms FQE) using models trained on data corresponding to the behavior policy $\pi_{b_2}$) and the 2DWorld environment

| Evaluation policy | FQE error on partition A | Model-based error on partition A | FQE error on partition B | Model-based error on partition B |
|---|---|---|---|---|
| $\pi_{b_2}$ | 5.9 | 27.0 | 19.9 | 7.7 |
| $\pi_{e_1}$ | 3.2 | 4.8 | 36.0 | 8.6 |
| $\pi_{e_2}$ | 7.8 | 18.0 | 18.5 | 7.5 |
| $\pi_{e_3}$ | 12.1 | 49.4 | 7.5 | 14.0 |

Table 7: Proportion of times method with the higher stability out of FQE and the model-based method had the higher error across pairs of behavior and evaluation policies using the 2DWorld environment

| Behavior policy | Evaluation policy | Proportion |
|---|---|---|
| $\pi_{b_1}$ | $\pi_{e_1}$ | 0.65 |
| $\pi_{b_1}$ | $\pi_{e_2}$ | 0.63 |
| $\pi_{b_1}$ | $\pi_{e_3}$ | 0.62 |
| $\pi_{b_1}$ | $\pi_{e_4}$ | 0.77 |
| $\pi_{b_2}$ | $\pi_{e_1}$ | 0.63 |
| $\pi_{b_2}$ | $\pi_{e_2}$ | 0.59 |
| $\pi_{b_2}$ | $\pi_{e_3}$ | 0.74 |
| $\pi_{b_2}$ | $\pi_{e_4}$ | 0.49 |

Table 8: Error summaries for conditional value estimates from $(s, a, r, s')$ tuples from monte carlo rollouts of the evaluation policy on the 2DWorld environment for perturbations on adaptive stability weighting methods and baseline methods.

| Method | Average absolute error |
|---|---|
| SSW | 9.2 |
| SSW 0.15/0.85 | 9.3 |
| SSW 0.25/0.75 | 9.6 |
| SSWPR | 9.2 |
| SSWPR 5 | 9.1 |
| SSWPR 7 | 8.9 |

Table 9: Mean absolute error for conditional value estimates from $(s, a, r, s')$ tuples from monte carlo rollouts of the evaluation policy on the mountain car task for proposed adaptive stability weighting methods and baseline methods)

| Evaluation policy | Behavior policy | FQE | Model-based | FQE and model-based average | Oracle non-adaptive selection | SSW | SSWPR |
|---|---|---|---|---|---|---|---|
| $\pi_{b_1}$ | $\pi_{b_1}$ | 6.8 | 9.6 | 7.6 | 6.8 | 5.6 | 5.4 |
| $\pi_{b_1}$ | $\pi_{e_1}$ | 14.0 | 3.1 | 5.9 | 3.1 | 3.5 | 3.5 |
| $\pi_{b_1}$ | $\pi_{e_2}$ | 7.4 | 5.0 | 5.9 | 5.0 | 5.6 | 5.5 |
| $\pi_{b_1}$ | $\pi_{e_3}$ | 12.1 | 5.2 | 8.2 | 6.2 | 5.0 | 5.9 |
| $\pi_{b_2}$ | $\pi_{b_2}$ | 6.1 | 5.0 | 4.8 | 5.0 | 4.8 | 4.8 |
| $\pi_{b_2}$ | $\pi_{e_1}$ | 16.3 | 14.7 | 15.3 | 14.7 | 13.9 | 12.6 |
| $\pi_{b_2}$ | $\pi_{e_2}$ | 7.6 | 7.4 | 7.3 | 7.4 | 7.2 | 6.7 |
| $\pi_{b_2}$ | $\pi_{e_3}$ | 15.5 | 6.8 | 10.8 | 6.8 | 6.6 | 6.3 |

## C    Additional results

### C.1    Empirical evidence for leveraging model stability across an ensemble

We first empirically show evidence that FQE and model-based evaluation can outperform one another conditioned on the state-space despite having been trained on the same logged data. We then show that stability from an ensemble of models initialized with different random seeds provides a positive signal for adaptive model weighting.

First, we compare FQE and model-based estimates on $(s, a, r, s')$ tuples deriving from the evaluation policy. We first sample rollouts from the evaluation policy on the actual environment and sample $500$ $(s, a, r, s')$ tuples from the resulting dataset. For each sample tuple, we extract the q-value from the trained FQE and dynamics models and the online target estimate using the true environment via monte carlo rollouts. Figures 5.6 and 5.7 show the sample tuples and highlights which method outperforms the other on the conditioned state. We see that empirically success in extrapolation to states with lower coverage is dependent on the behavior policy and evaluation policy pair and the state itself. Tables 5 and 6 show that the difference in errors of each method on partitions where one outperforms the other is large. This suggests that weighting between FQE and the model-based approach may considerably improve value estimation averaged across samples across the state-space if we are able to determine which method is likely to outperform the other adaptive to the state the agent is on.

Next, we show how stability via ensembling models is one viable avenue for extracting a positive signal for local predictiveness, which is related to weighted online learning. However unlike previous work which uses local predictiveness as a weighting mechanism, we use the stability of models. First we train an ensemble of $5$ FQE models and $5$ pairs of reward and transition models using a random initialization of the network weights using the same procedure outlined above. For each tuple in the sampled tuples deriving from the evaluation policy, we then compute the stability from the ensemble, defined by the standard deviation across the ensemble predictions. We report the proportion of times the method with more instability had the higher error conditioned on the state and action pair from the sampled tuples from the evaluation policy in Table 5.4. The high proportions across evaluation and behavior policy pairs show the positive relation between model stability and the error of the model in this offline evaluation setting. Note that this signal, although close to $0.5$ for some pairs, is powerful in this setting where it is not possible to estimate model errors on a hold-out set, and thus do any type of model selection or tuning.

**Ablation to algorithm hyperparameters: Stability of results on human decisions**

We additionally test the stability of the results of SSW and SSWPR against ad-hoc decisions for the parameters. The first involves perturbing the soft weighting normalization outlined previously. Note that the following equation contains a human decision of using the $0.2$ and $0.8$ quantiles of the distribution of uncertainties stemming from the behavior dataset:

$$u_{FQE}(s, a) = \frac{\hat{\sigma}_{FQE}(s, a) - \text{quantile}_{\mathbf{D}_{\pi_b}}(\hat{\sigma}_{FQE}, 0.2)}{\text{quantile}_{\mathbf{D}_{\pi_b}}(\hat{\sigma}_{FQE}, 0.8) - \text{quantile}_{\mathbf{D}_{\pi_b}}(\hat{\sigma}_{FQE}, 0.2)}.$$

We check whether the results are robust across perturbations of the quantiles used. On this end we use two pairs of values $(0.15, 0.85)$ and $(0.25, 0.75)$ and rerun the SSW. Results are shown in table 5.7 and summarized in table 5.8.

We next check the stability of the SSWPR method against the decision of using max horizon length of $5$ when creating partial rollouts. We use values of horizon lengths of $3$ and $7$ and rerun SSWPR. Results are shown in table 5.7 and summarized in table 5.8. Note that on average deviations from the original settings of SSW and SUPRW are small, suggesting that the two methods are robust to these parameters. We even see that, on average, both perturbations to SSWPR perform marginally than the original SSWPR (8.9/9.1 vs 9.2 average errors). Thus, the results in table 5.5 and 5.6 do not seem to rely on the particular choices of parameters used.

### C.2 Additional details for mountain car task

Note that the set up and hyperparameters are kept the same mostly the same as the experiments on the 2D world environment. One difference is the stopping condition for FQE was changed to be

$$|TD_{i-5,i} - TD_{i-6,i-1}| < 0.00025,$$

instead of using $0.001$ like in 2DWorld. The behavior policies $\pi_{b_1}$ and $\pi_{b_2}$ were trained and achieve mean values of $-82.5$ and $-82.8$, respectively. The evaluation policies $\pi_{e_1}$, $\pi_{e_2}$, and $\pi_{e_3}$ achieved mean values of $-95.1$, $-74.5$, and $-75.9$, respectively.

Similar to the 2DWorld task, we visualize individual predictions of SSW, SSWPR, FQE, and the model-based method against the target values for the mountain car task in figs. 22 to 25. We include histogram of absolute errors of SSW, SSWPR, FQE, and the model-based method in Figures 26 to 29, 31 and 32 and appendix D. Like the results on the 2DWorld task, SSW and SSWPR tend to reduce the extremity of outlier estimates produced by the model-based method or FQE on average. We also include scatterplots comparing predictions between SSW and SSWPR for the mountain car task in figs. 33 and 34 which show that the predictions between the two methods are very similar. However, there are cases where moderate variation exist between two methods as shown in parts C and D in figs. 33 and 34 which imply one may outperform over the other.

## Discussion

We have investigated using stability across an ensemble of neural networks with different initializations as a weighting mechanism in the setting of offline evaluation of reinforcement learning agents and have proposed two methods, SSW and SSWPR, to incorporate stability for adaptively combining conditional model estimates. Our methods provide a positive signal for when a particular method is suited given a local region of the state space. The guiding principle behind the two methods is that if a model's prediction is unstable then its predictive estimate should not be trusted. This principle is powerful in the offline setting where it is unclear how to validate a model's prediction and compare against other models. These methods are particularly valuable in the offline evaluation setting, because a variety of algorithms exist for providing value estimates while model selection is an open problem.

We test our methods on three simulated environments across combinations of different behavior and evaluation policies. By leveraging stability values stemming from model ensembling, we are able to outperform one of FQE and the model-based method every time and both FQE and the model-based half the time. Our experiments suggest that using stability can provide improvements when used to combine estimates of different evaluation algorithms.

SSW and SSWPR are related to the idea of CLEP ensembling [1] in weighted online learning. CLEP produces a weighted average of predictions from individual time series models where the weight of a model's prediction given a set of features is based on the recent performance of the predictor on past data. Unlike the covid-19 forecasting setting, our offline evaluation models do not have comparable performance metrics due to the distributional shift between the behavior and evaluation policies. Thus, we rely on the stability of the model to extract a signal about local predictability. Our experiments show that stability is empirically correlated with the error of the model.

We note that both SSW and SSWPR are related to the idea of perturbation intervals outlined in the PCS framework which quantify the stability of target estimates to perturbations [15]. In our setting, the perturbations are at the data and model level, where randomization of the model initialization is used as perturbations. The notion underlying both methods are that the resulting variability in estimates outline regions of the state space which models are stable and unstable. The key assumption is that stability to such perturbations can help to identify in which scenarios a certain model or method is more reliable than another.

Our proposed methods SSW and SSWPR are also related to the pessimism principle studied in offline reinforcement policy learning. The main notion of the pessimism principle leveraged in these works is that areas of the state-space where the Q-function or dynamics model is uncertain or unstable should be avoided due to insufficient coverage of the behavior policy. Kidambi et al. [7]

incorporate pessimism into model-based reinforcement learning and construct a pessimistic markov decision process using an ensemble of learned dynamics models which partitions the state space into known and unknown regions and artificially penalizes an agent with a negative reward for visiting unknown regions. Kumar et al. [10] train policies by maximizing the most conservative estimate from an ensemble of Q-functions as well constraining the policy to the support of the behavior policy

$$\max_{\pi \in \prod_\epsilon} \mathbb{E}_{a \sim \pi(.|s)} \left[ \min_{j=1...K} \hat{Q}_j(s,a) \right],$$

where $\prod_\epsilon$ is the set of policies sharing the support of the behavior policy. We apply similar reasoning to the offline evaluation setting by using the stability of model ensembles to weight offline estimates.

Our study has a few limitations that should be mentioned. First, we have only provided empirical evidence for using stability as a model weighting mechanism across a few simulated environments. Further experimentation and testing needs to be done to better understand the soft weighting mechanism and under which cases the soft weighting is likely to lead to significant improvements in performance over baselines as well as benchmarking on more complex environments. Our experiments show that if the bias of FQE and the model-based method are in opposite directions, improvements over both FQE and the model-based method are likely. Additionally, our weighting methods outperform a simple average of FQE and the model-based method, which show the utility in using stability as a weighting mechanism over naive averaging. A second limitation is that our methods are based on parameters including the weighting normalization for SSW and the partial horizon length for SSWPR which may require tuning based on the specific application. Although our experiments shows that SSW and SSWPR are robust to perturbations of the values for the parameters used in the environments we test on, more experimentation should be done. Lastly, we use perturbations on the model level via different randomization of neural network weights to evaluate stability. However, other forms of perturbations can and should be experimented, including those at the data level, which can be challenging in the reinforcement learning setting. One future direction is to include bootstrapped data as an avenue of assessing stability.

## CONCLUSION

Model selection and validation is a difficult problem in the offline reinforcement learning setting due to the lack of access to the environment for data collection. We propose using stability of evaluation functions as a weighting mechanism inspired by ideas from weighted online learning when typical validation is not possible. Our proposed methods SSW and SSWPR, which leverage stability via ensembling, have shown to improvement offline estimates from FQE and the model-based method, potentially increasing the viability of reinforcement learning to applications like healthcare where safety is paramount.

 # D    Experimental figures

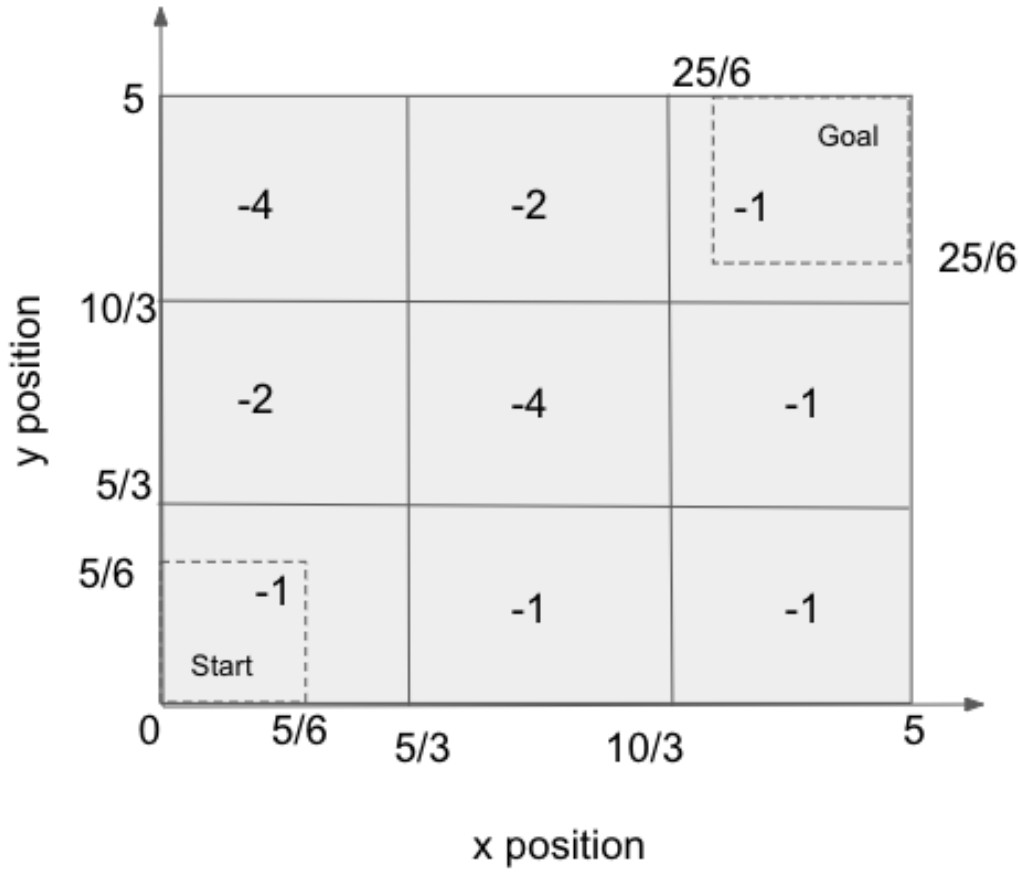

Figure 1: 2DWorld: positional depiction. Agent starts at a random position from $[0, 5/6]$ in both its $x$ and $y$ positions and receives a negative reward at each time step conditioned on the subgrid the agent currently subsides. The agent receives a completion reward of $+10$ if the goal is reached ($[25/6, 5]$ in both its $x$ and $y$ positions.)

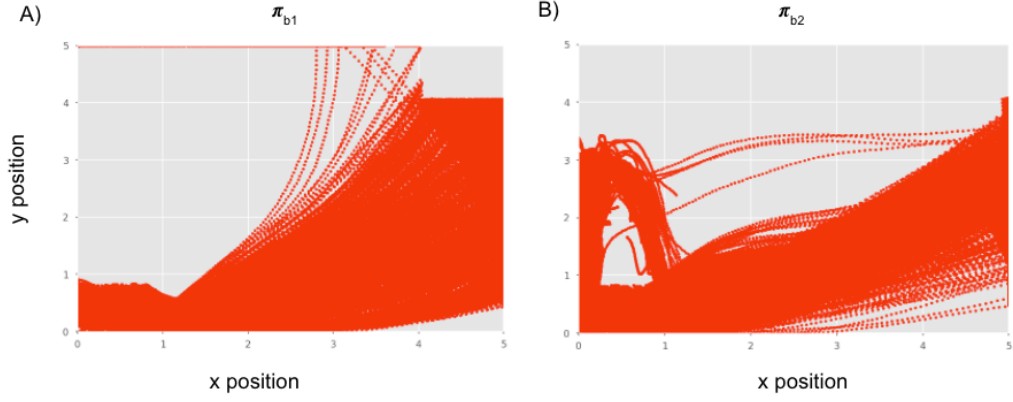

Figure 2: Vertical and horizontal positions over time from episodes derived from behavior policies $\pi_{b1}$ (A) and $\pi_{b2}$ (B) trained on the 2DWorld environment

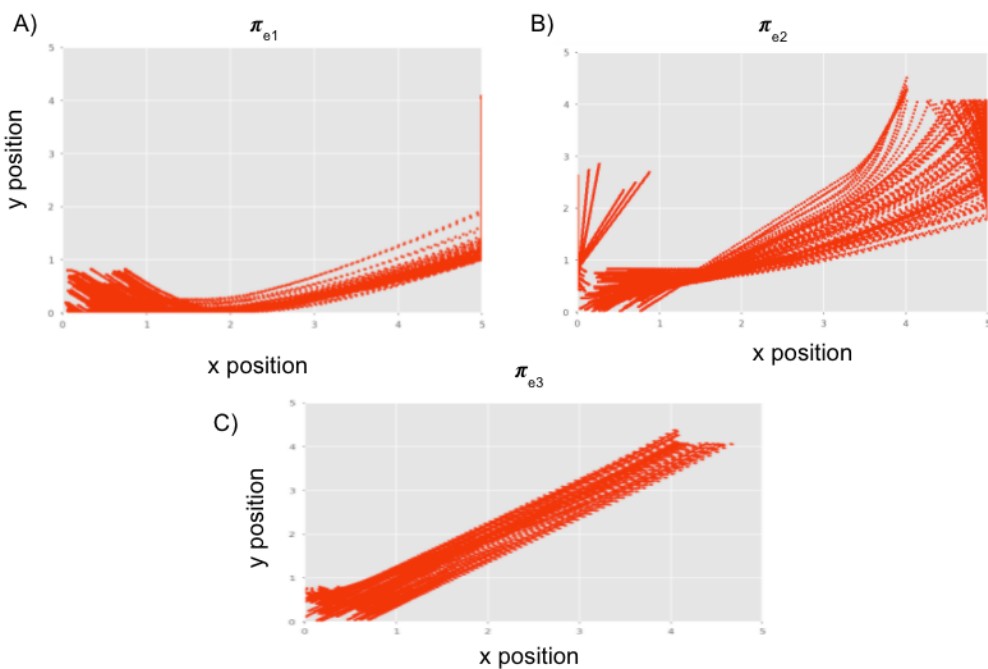

Figure 3: Vertical and horizontal positions over time from episodes derived from evaluation policies $\pi_{e1}$ (A), $\pi_{e2}$ (B), and $\pi_{e3}$ (C) trained on the 2DWorld environment

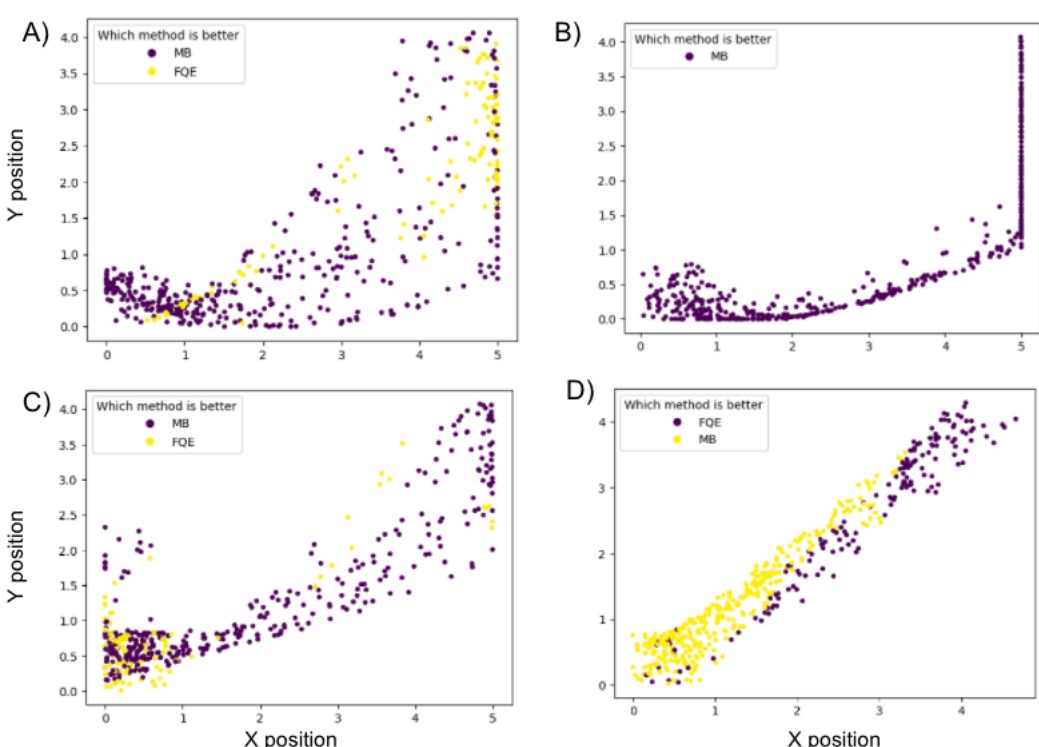

Figure 4: Scatterplots of the x and y positions of sampled tuples on the 2DWorld environment colored by the best performing method trained on the behavior dataset corresponding to $\pi_{b_1}$. The evaluation policies used include $\pi_{b_1}$ (A), $\pi_{e_1}$ (B), $\pi_{e_2}$ (C), $\pi_{e_3}$ (D).

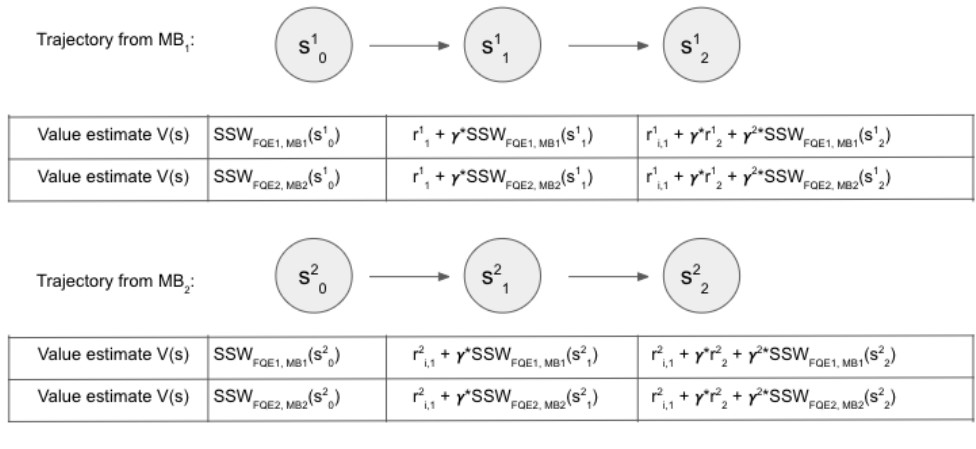

SSWPR $(s_0)$ = $p(s_0)$*median($V(s_0)$) + $p(s_1)$*median($V(s_1)$) + $p(s_2)$*median($V(s_2)$)

where $p(s_j)$ = softmax(-std_dev($s_0$), -std_dev($s_1$), -std_dev($s_2$))$_j$

Figure 5: A depiction of the SSWPR method with ensembles of size two for FQE and the model-based method and a maximum partial horizon of 3 where std_dev represents the standard deviation, median($V(s_t)$) represents the median of all value estimates at time $t$, $s_j^i$ represents the simulated state from dynamics model $i$ at time step $j$, and $r_j^i$ represents the reward given from dynamics model $i$ at time step $j$.

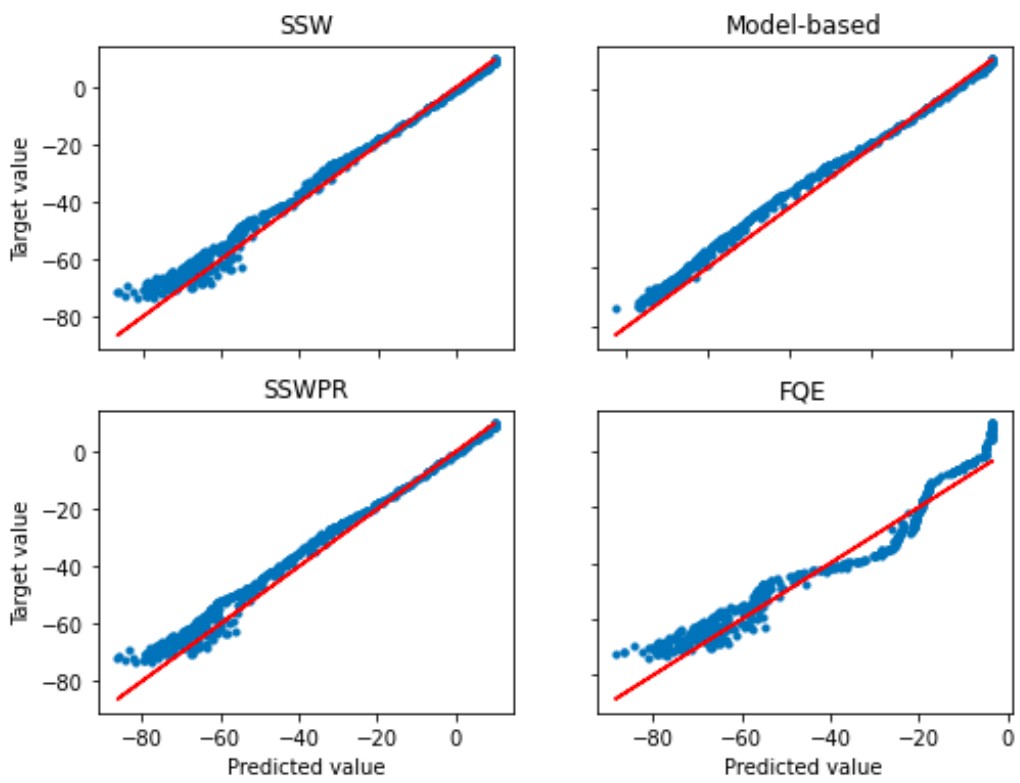

Figure 7: Scatterplots of the predictions and target values for evaluating $\pi_{e_1}$ on the logged data deriving from $\pi_{b_1}$ on the 2DWorld environment using the model-based method, FQE, SSW, and SSWPR.

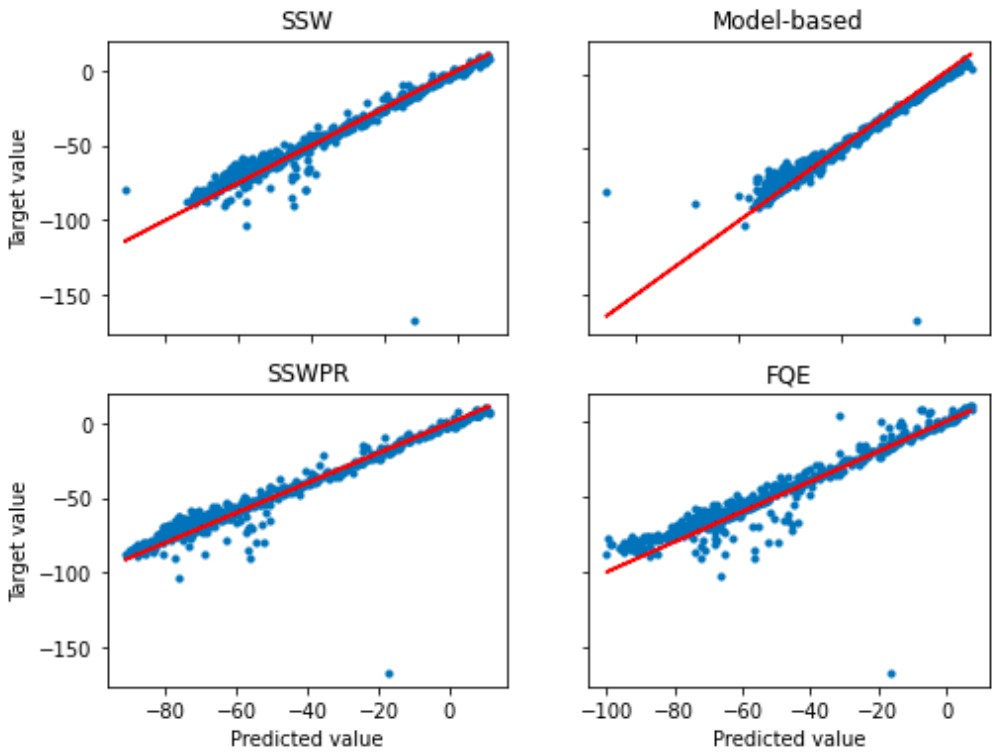

Figure 6: Scatterplots of the predictions and target values for evaluating $\pi_{b_1}$ on the logged data deriving from $\pi_{b_1}$ on the 2DWorld environment using the model-based method, FQE, SSW, and SSWPR.

Figure 8: Scatterplots of the predictions and target values for evaluating $\pi_{e_2}$ on the logged data deriving from $\pi_{b_1}$ on the 2DWorld environment using the model-based method, FQE, SSW, and

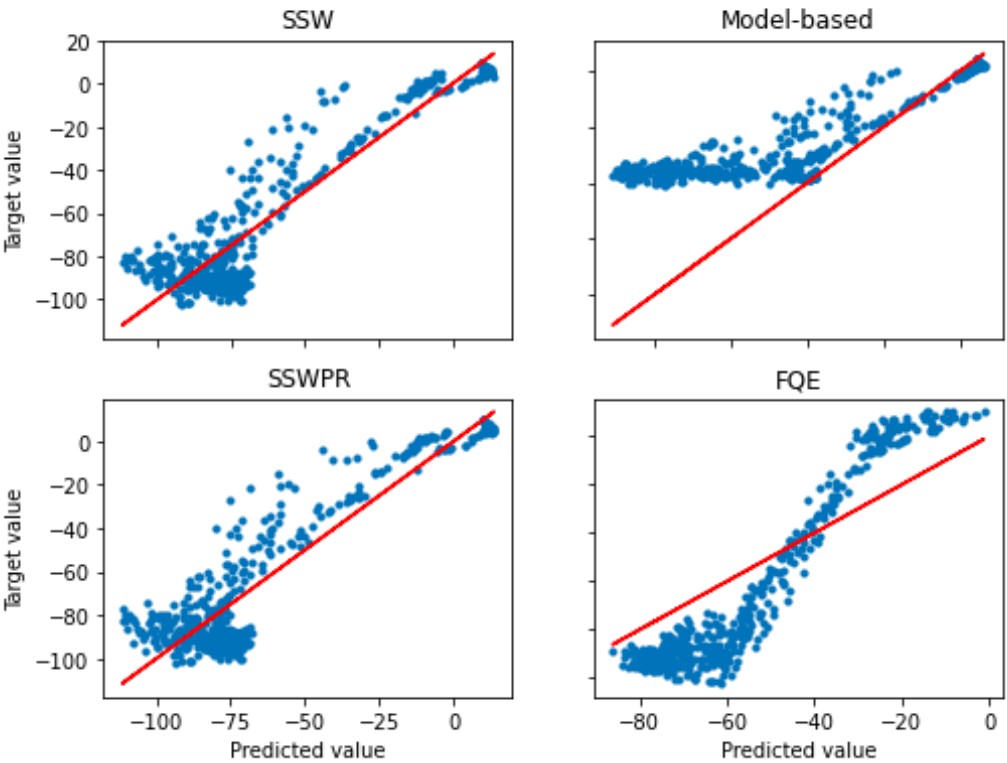

Figure 9: Scatterplots of the predictions and target values for evaluating $\pi_{e_3}$ on the logged data deriving from $\pi_{b_1}$ on the 2DWorld environment using the model-based method, FQE, SSW, and SSWPR.

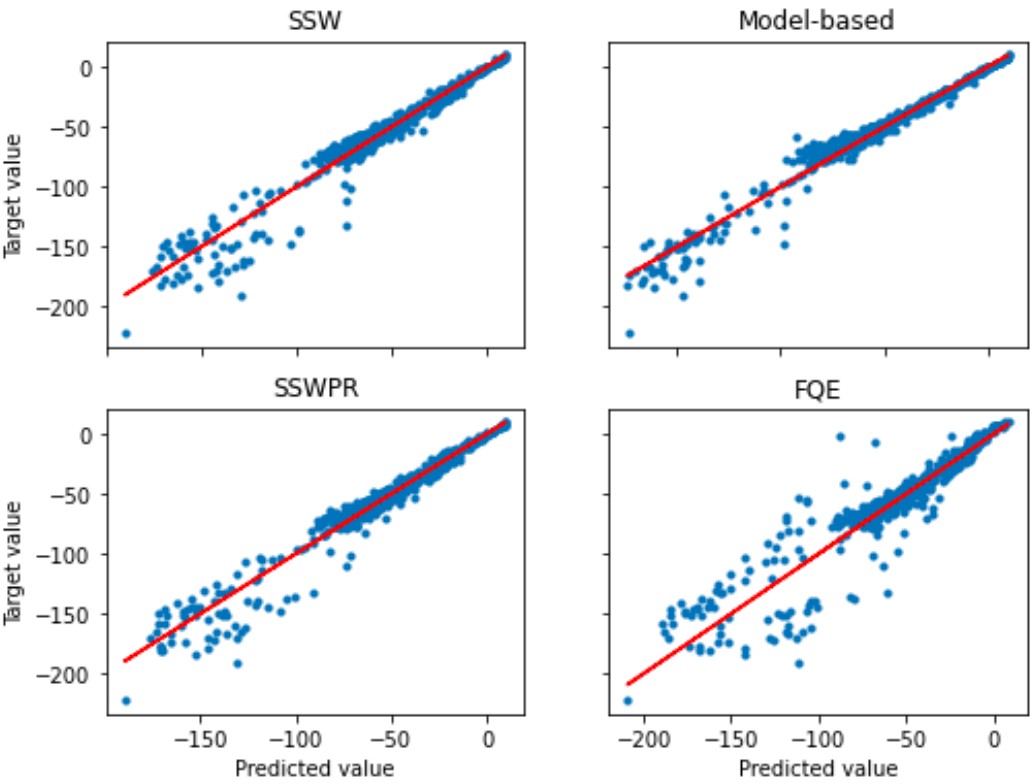

Figure 10: Scatterplots of the predictions and target values for evaluating $\pi_{b_2}$ on the logged data deriving from $\pi_{b_2}$ on the 2DWorld environment using the model-based method, FQE, SSW, and SSWPR.

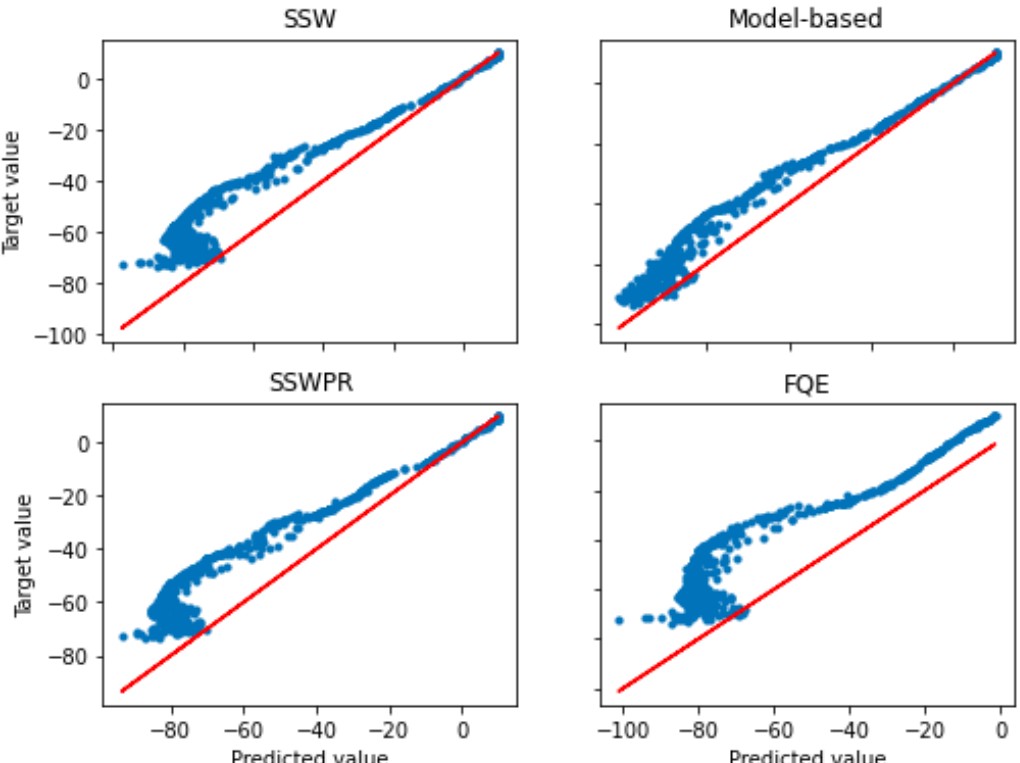

Figure 11: Scatterplots of the predictions and target values for evaluating $\pi_{e_1}$ on the logged data deriving from $\pi_{b_2}$ on the 2DWorld environment using the model-based method, FQE, SSW, and SSWPR.

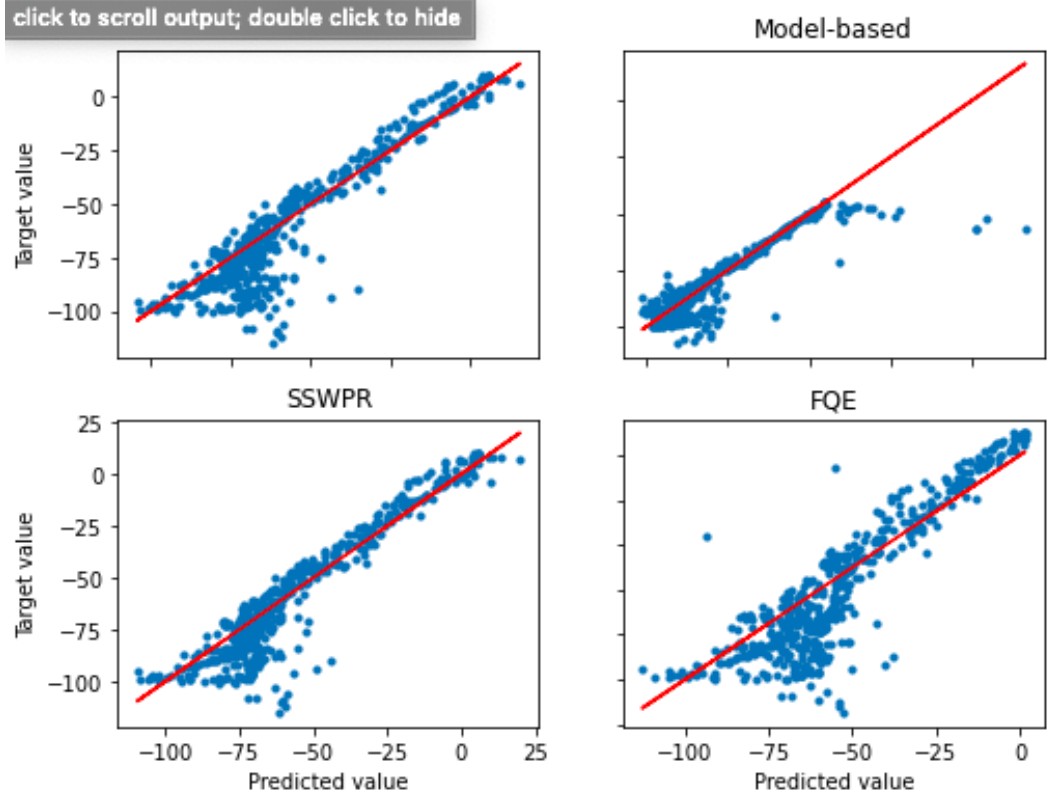

Figure 12: Scatterplots of the predictions and target values for evaluating $\pi_{e_2}$ on the logged data deriving from $\pi_{b_2}$ on the 2DWorld environment using the model-based method, FQE, SSW, and SSWPR.

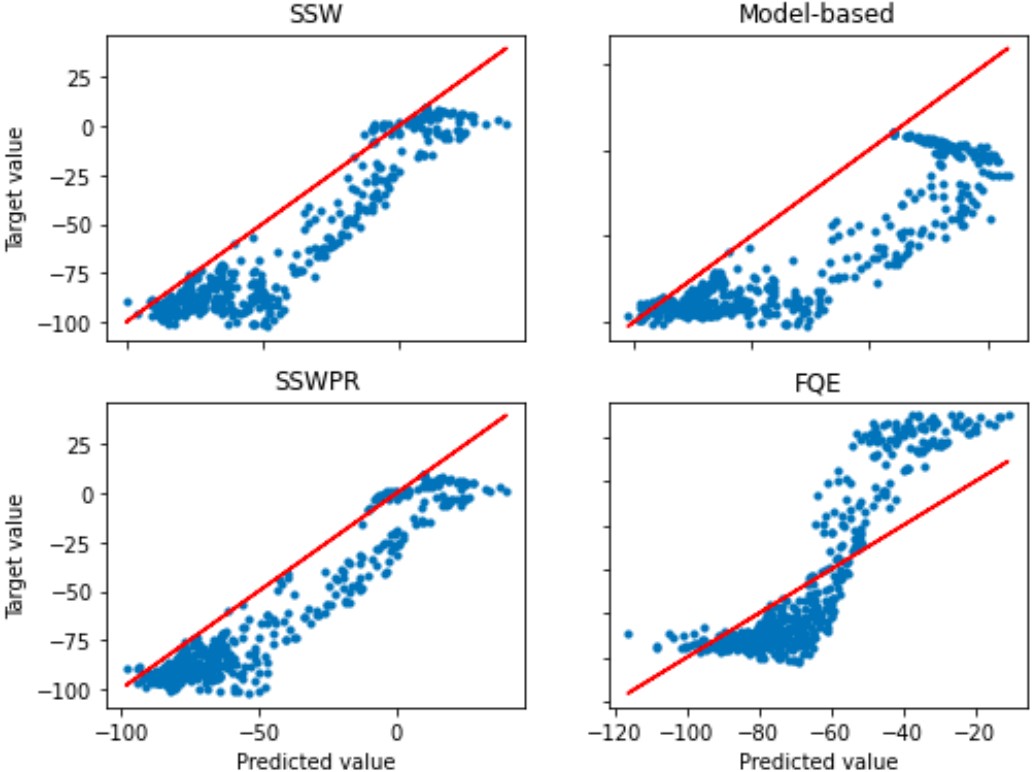

Figure 13: Scatterplots of the predictions and target values for evaluating $\pi_{e_3}$ on the logged data deriving from $\pi_{b_2}$ on the 2DWorld environment using the model-based method, FQE, SSW, and SSWPR.

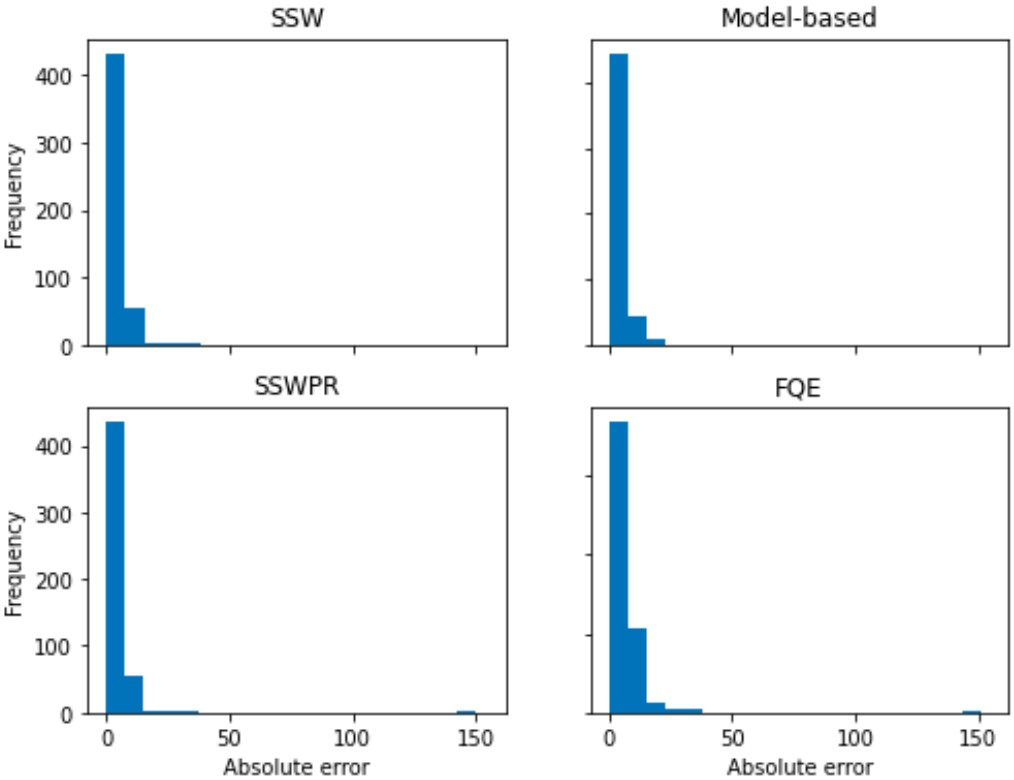

Figure 14: Histogram of absolute errors of SSW, SSWPR, FQE, and the model-based method for evaluating $\pi_{b_1}$ on the logged data deriving from $\pi_{b_1}$ on the 2DWorld environment

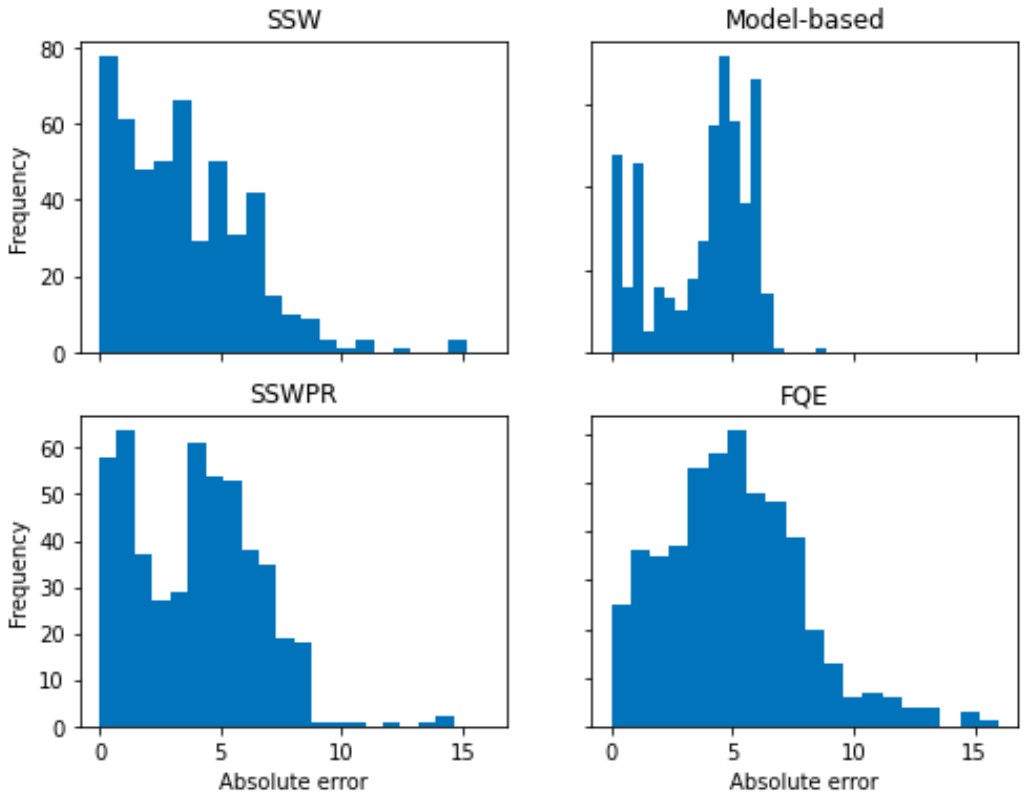

Figure 15: Histogram of absolute errors of SSW, SSWPR, FQE, and the model-based method for evaluating $\pi_{e_1}$ on the logged data deriving from $\pi_{b_1}$ on the 2DWorld environment

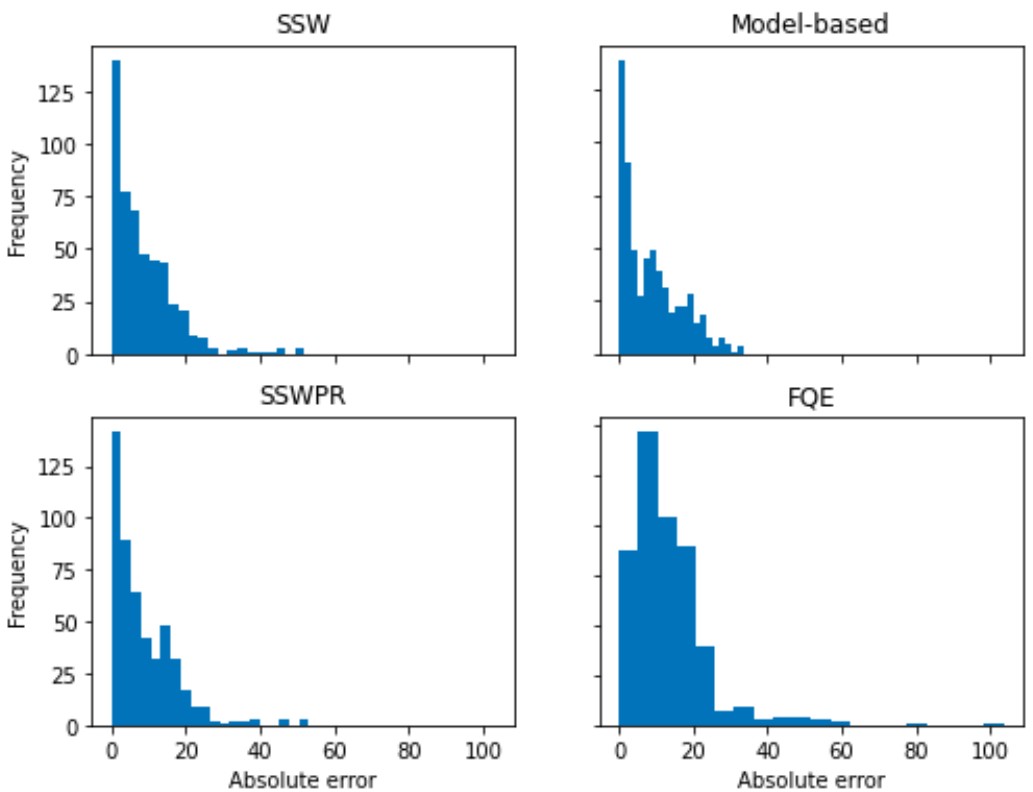

Figure 16: Histogram of absolute errors of SSW, SSWPR, FQE, and the model-based method for evaluating $\pi_{e_2}$ on the logged data deriving from $\pi_{b_1}$ on the 2DWorld environment

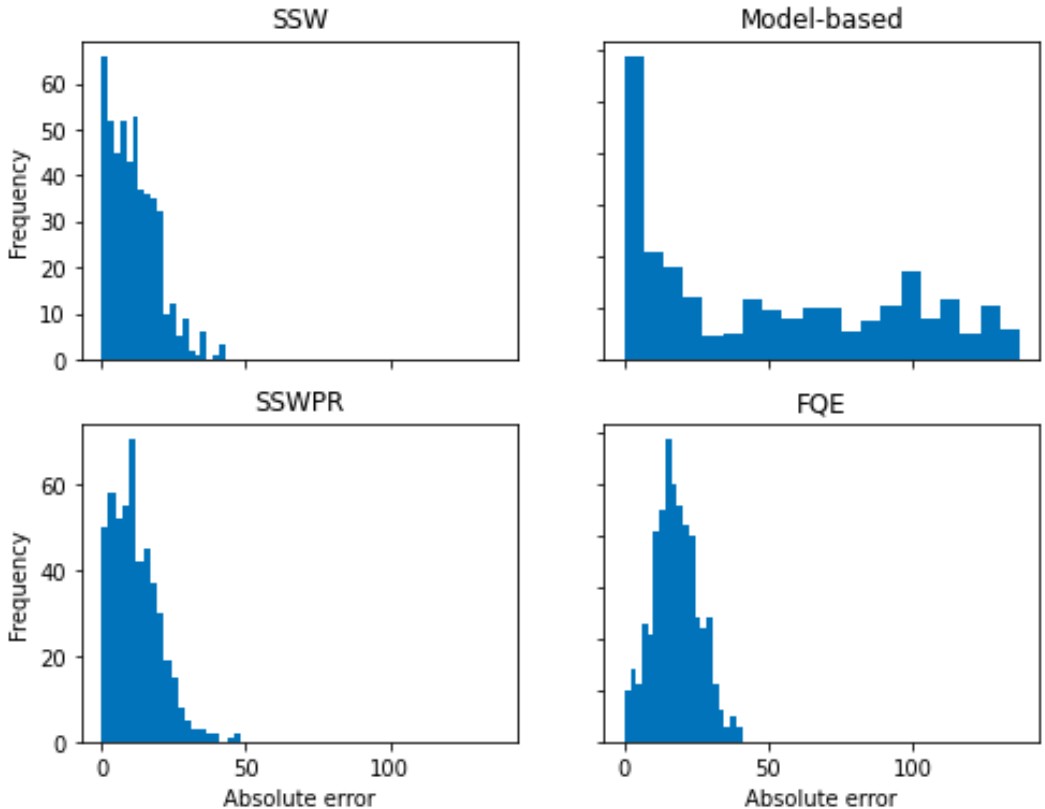

Figure 17: Histogram of absolute errors of SSW, SSWPR, FQE, and the model-based method for evaluating $\pi_{e_3}$ on the logged data deriving from $\pi_{b_1}$ on the 2DWorld environment

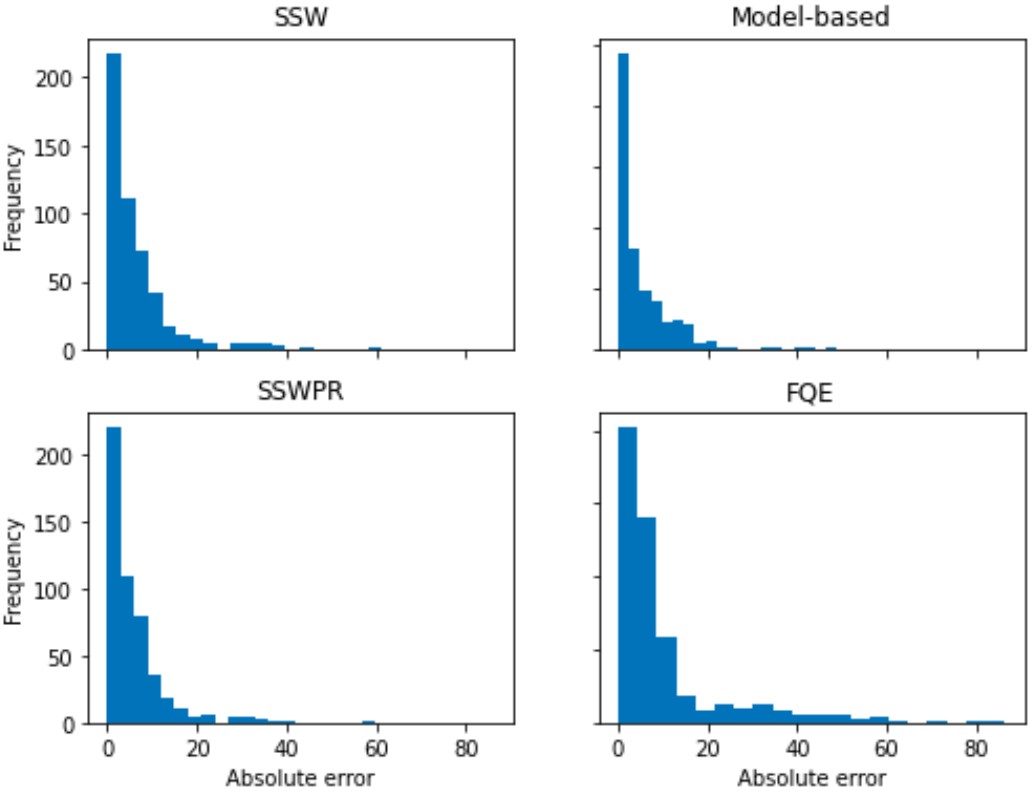

Figure 18: Histogram of absolute errors of SSW, SSWPR, FQE, and the model-based method for evaluating $\pi_{b_2}$ on the logged data deriving from $\pi_{b_2}$ on the 2DWorld environment

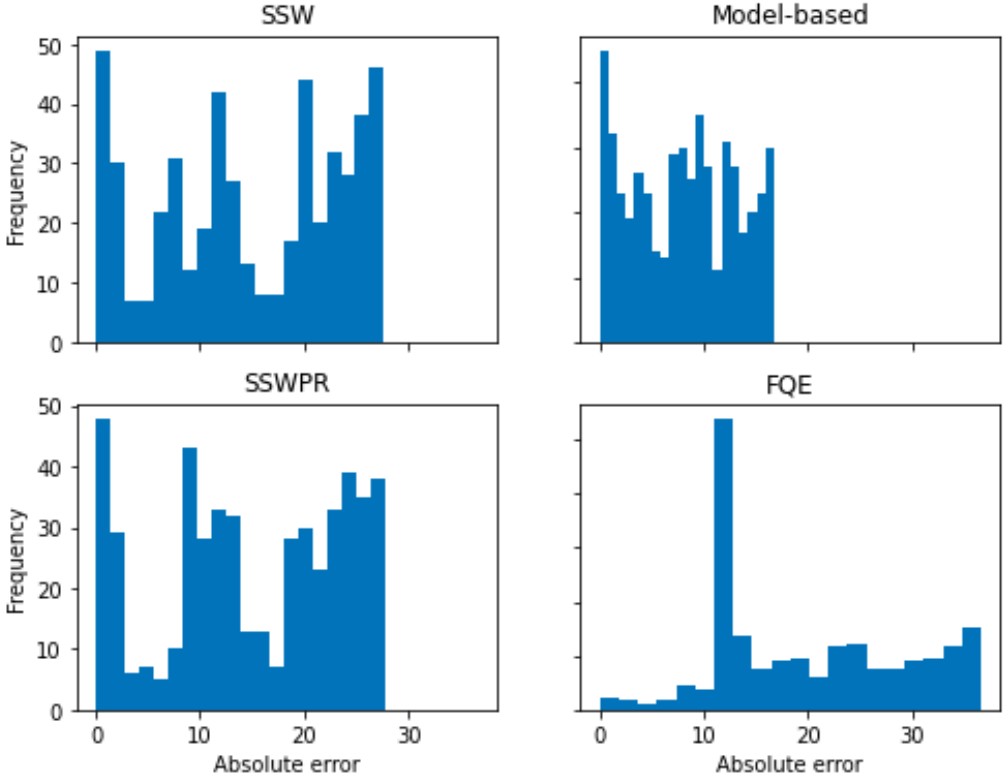

Figure 19: Histogram of absolute errors of SSW, SSWPR, FQE, and the model-based method for evaluating $\pi_{e_1}$ on the logged data deriving from $\pi_{b_2}$ on the 2DWorld environment

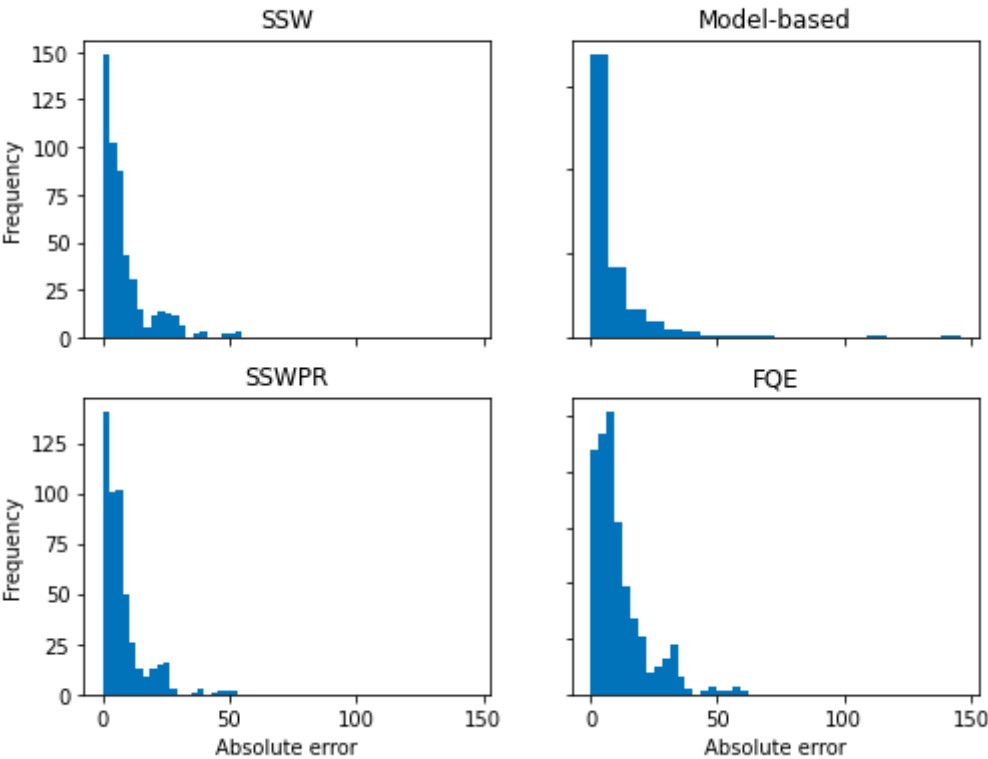

Figure 20: Histogram of absolute errors of SSW, SSWPR, FQE, and the model-based method for evaluating $\pi_{e_2}$ on the logged data deriving from $\pi_{b_2}$ on the 2DWorld environment

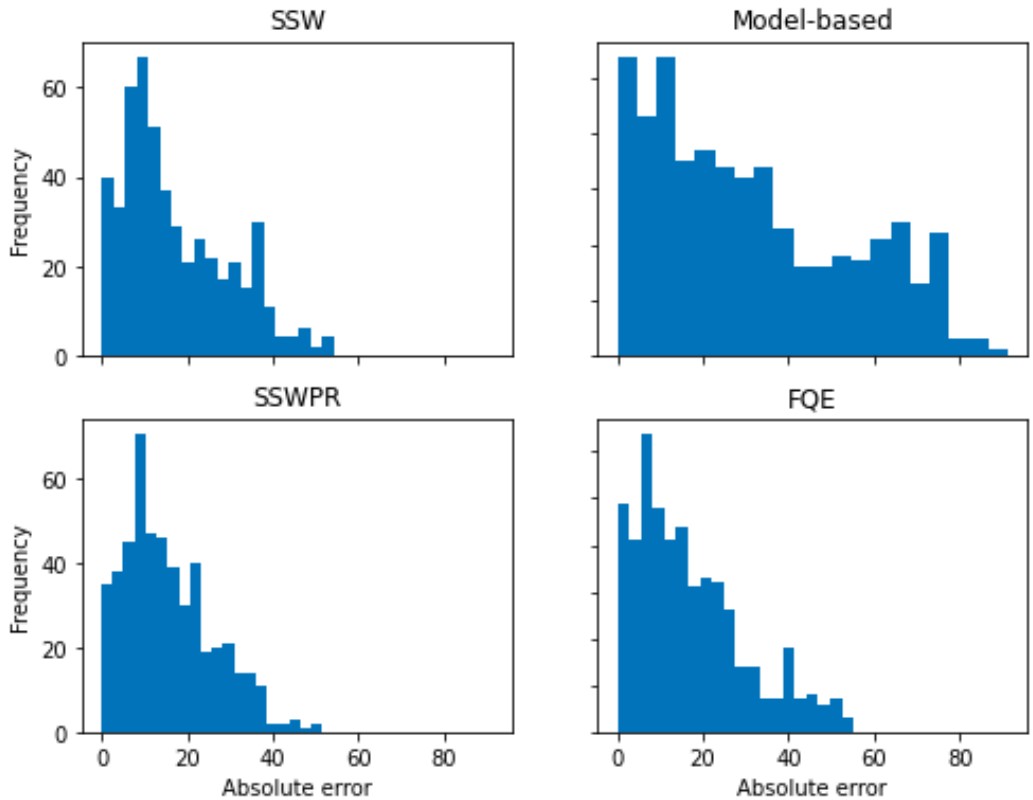

Figure 21: Histogram of absolute errors of SSW, SSWPR, FQE, and the model-based method for evaluating $\pi_{e_3}$ on the logged data deriving from $\pi_{b_2}$ on the 2DWorld environment

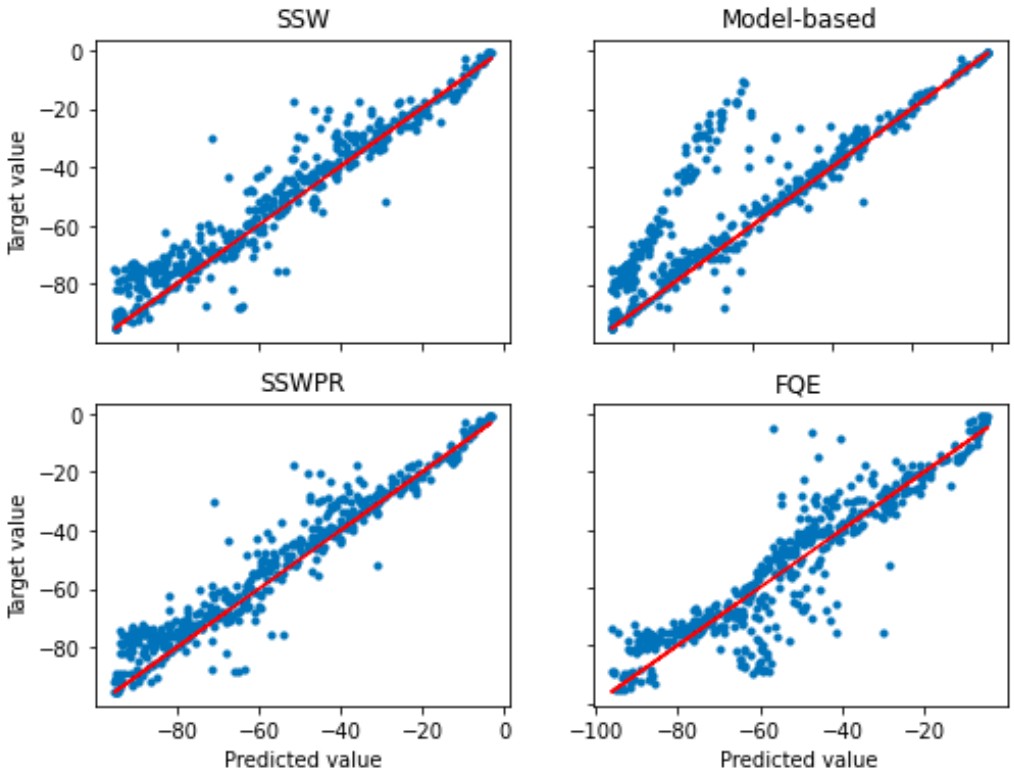

Figure 22: Scatterplots of the predictions and target values for evaluating $\pi_{b_1}$ on the logged data deriving from $\pi_{b_1}$ on the mountain car environment using the model-based method, FQE, SSW, and SSWPR.

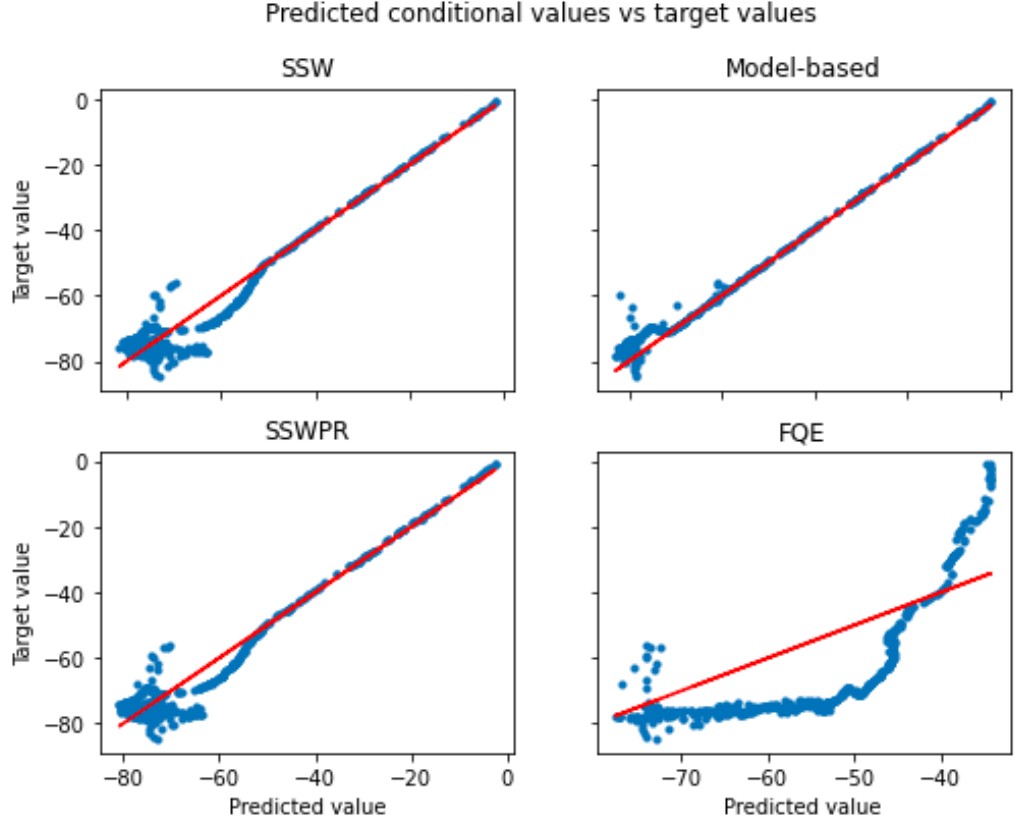

Figure 23: Scatterplots of the predictions and target values for evaluating $\pi_{e_1}$ on the logged data deriving from $\pi_{b_1}$ on the mountain car environment using the model-based method, FQE, SSW, and SSWPR.

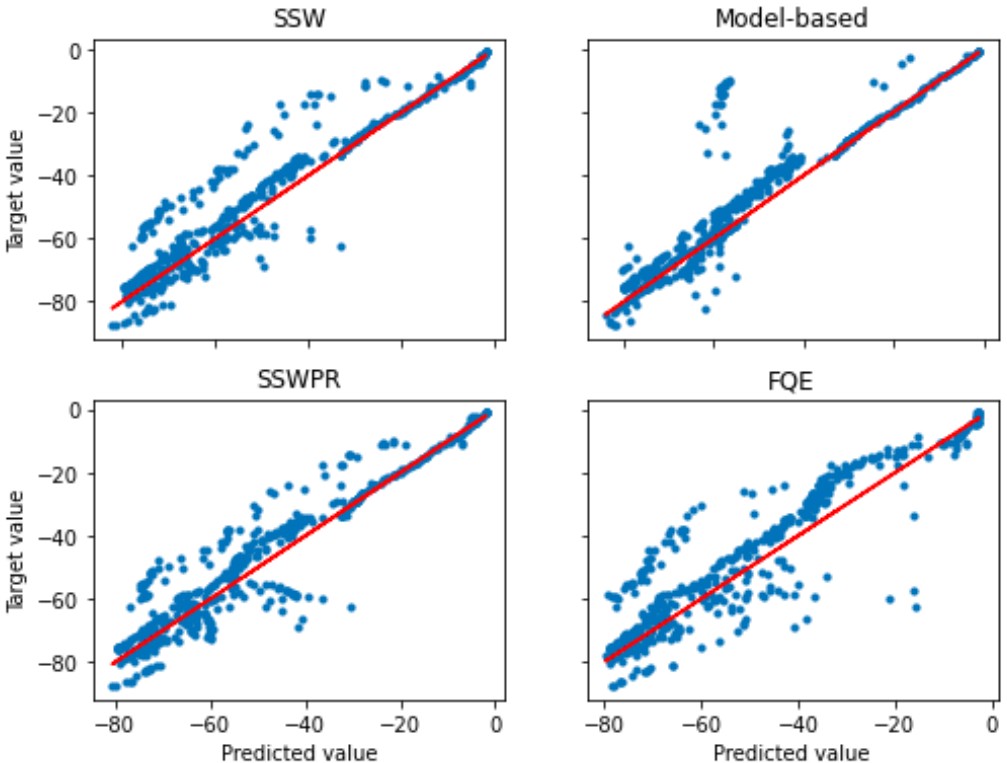

Figure 24: Scatterplots of the predictions and target values for evaluating $\pi_{e_2}$ on the logged data deriving from $\pi_{b_1}$ on the mountain car environment using the model-based method, FQE, SSW, and SSWPR.

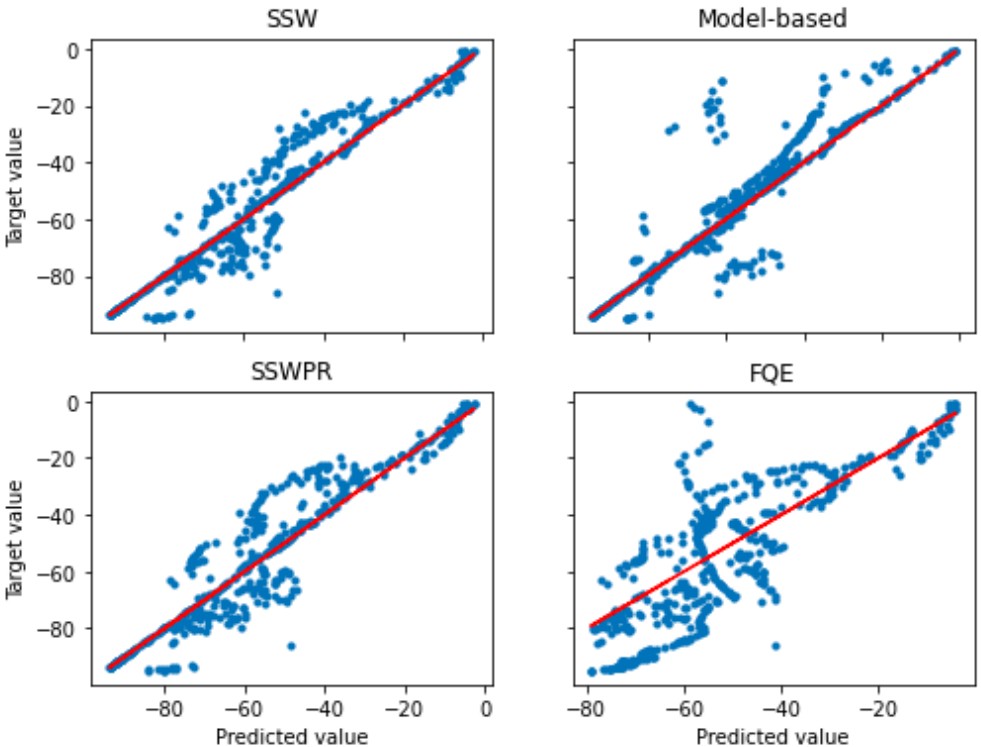

Figure 25: Scatterplots of the predictions and target values for evaluating $\pi_{e_3}$ on the logged data deriving from $\pi_{b_1}$ on the mountain car environment using the model-based method, FQE, SSW, and SSWPR.

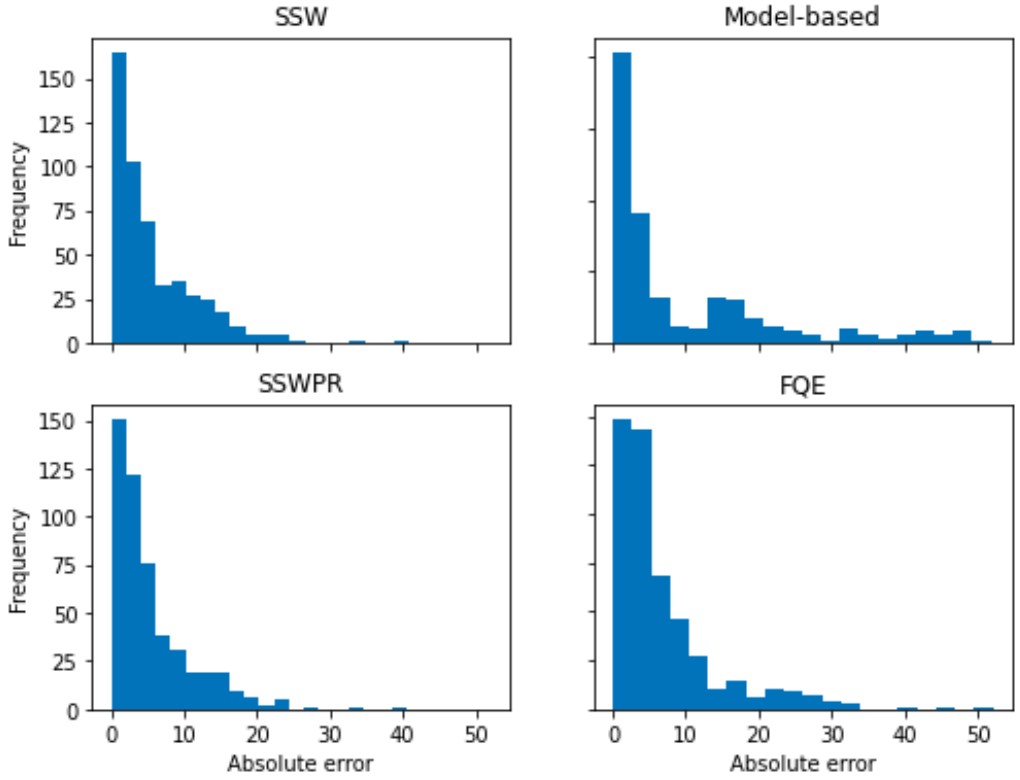

Figure 26: Histogram of absolute errors of SSW, SSWPR, FQE, and the model-based method for evaluating $\pi_{b_1}$ on the logged data deriving from $\pi_{b_1}$ on the mountain car environment

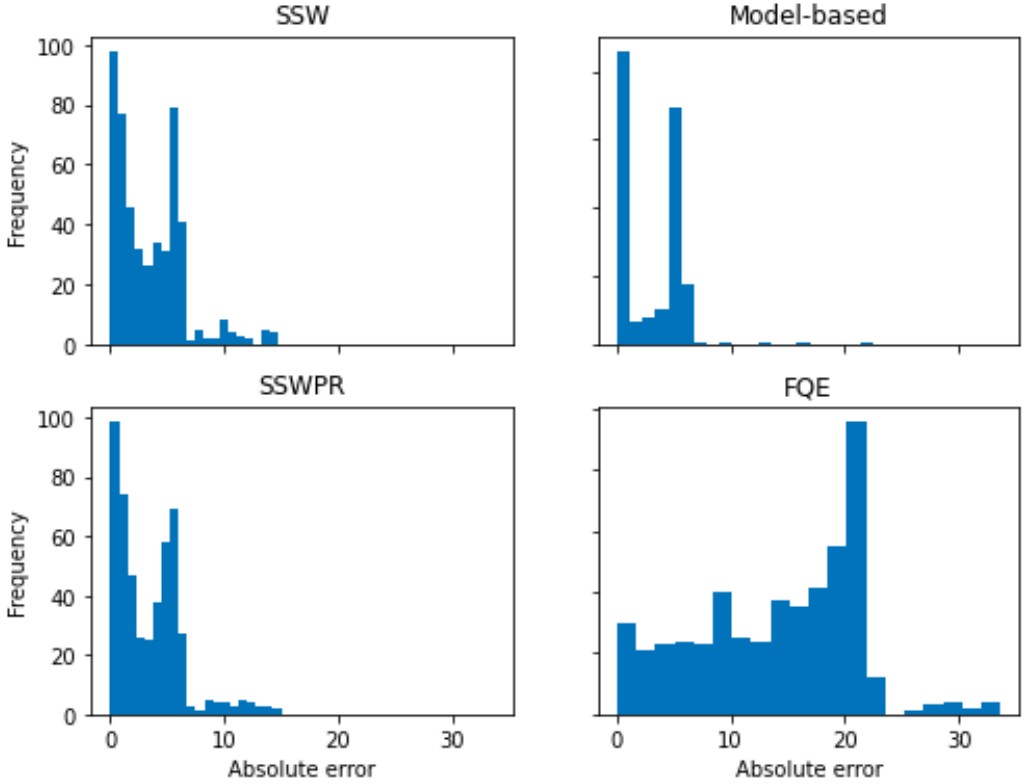

Figure 27: Histogram of absolute errors of SSW, SSWPR, FQE, and the model-based method for evaluating $\pi_{e_1}$ on the logged data deriving from $\pi_{b_1}$ on the mountain car environment

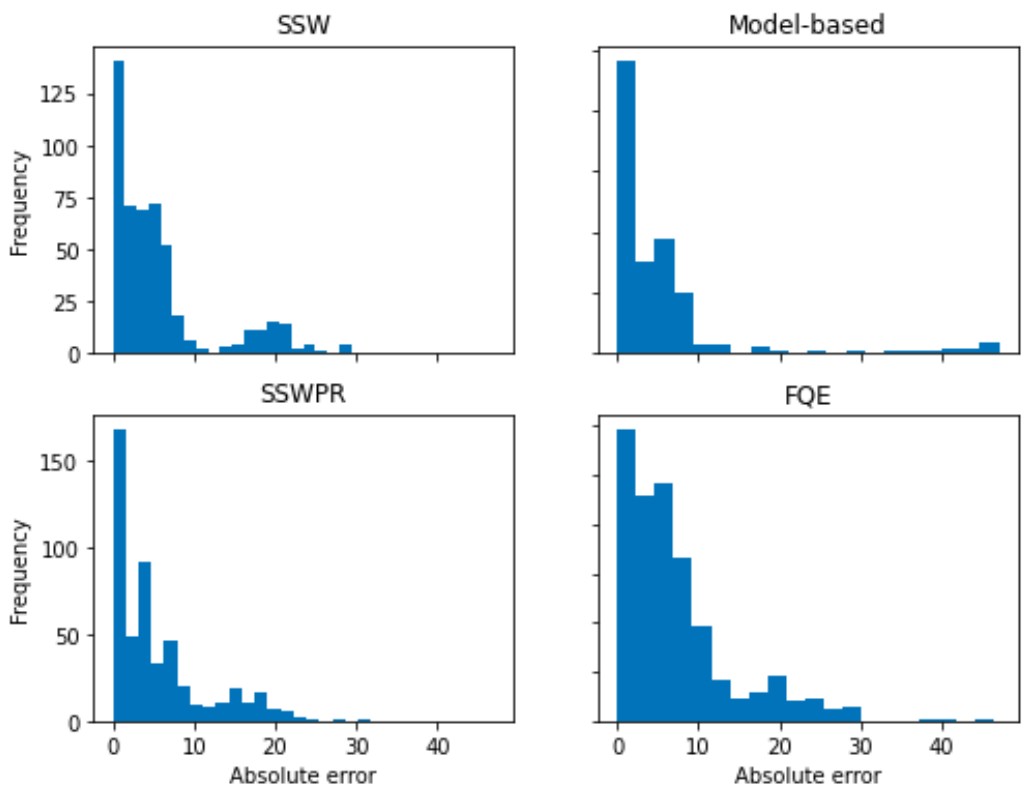

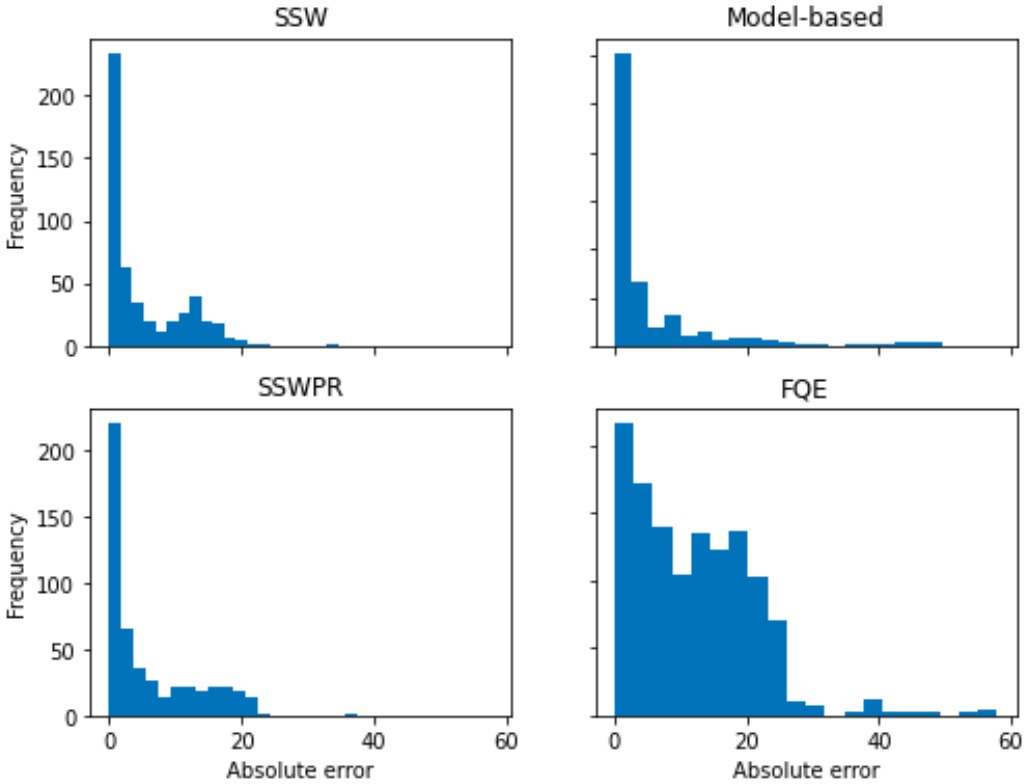

Figure 28: Histogram of absolute errors of SSW, SSWPR, FQE, and the model-based method for evaluating $\pi_{e_3}$ on the logged data deriving from $\pi_{b_1}$ on the mountain car environment

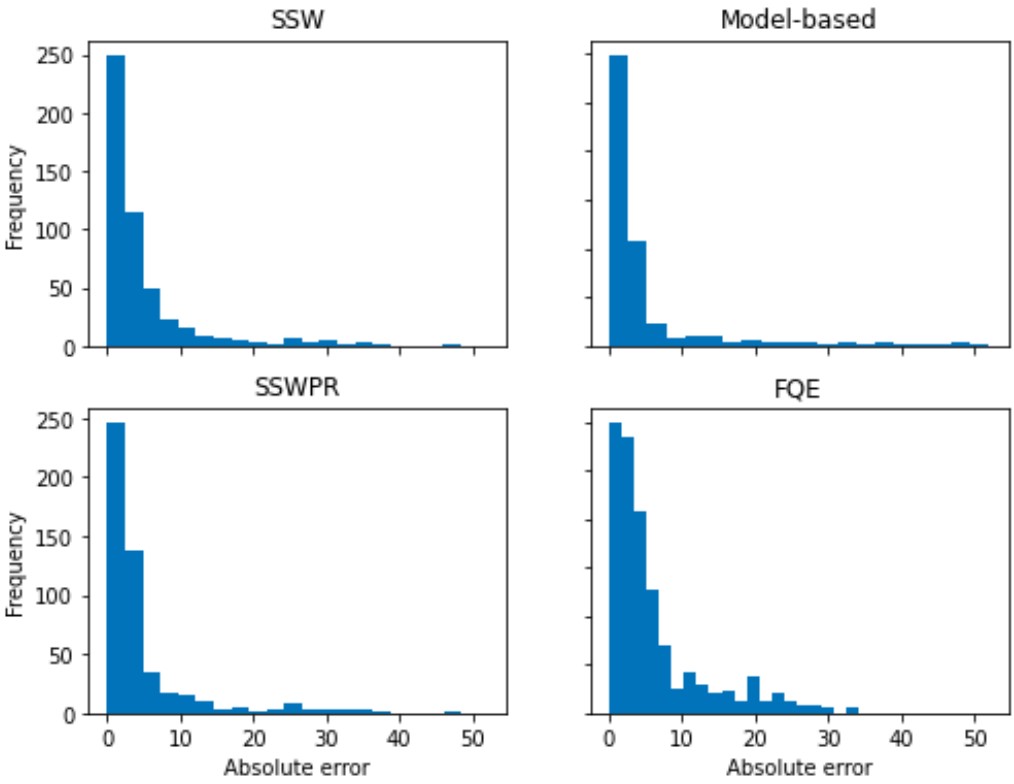

Figure 29: Histogram of absolute errors of SSW, SSWPR, FQE, and the model-based method for evaluating $\pi_{b_2}$ on the logged data deriving from $\pi_{b_2}$ on the mountain car environment

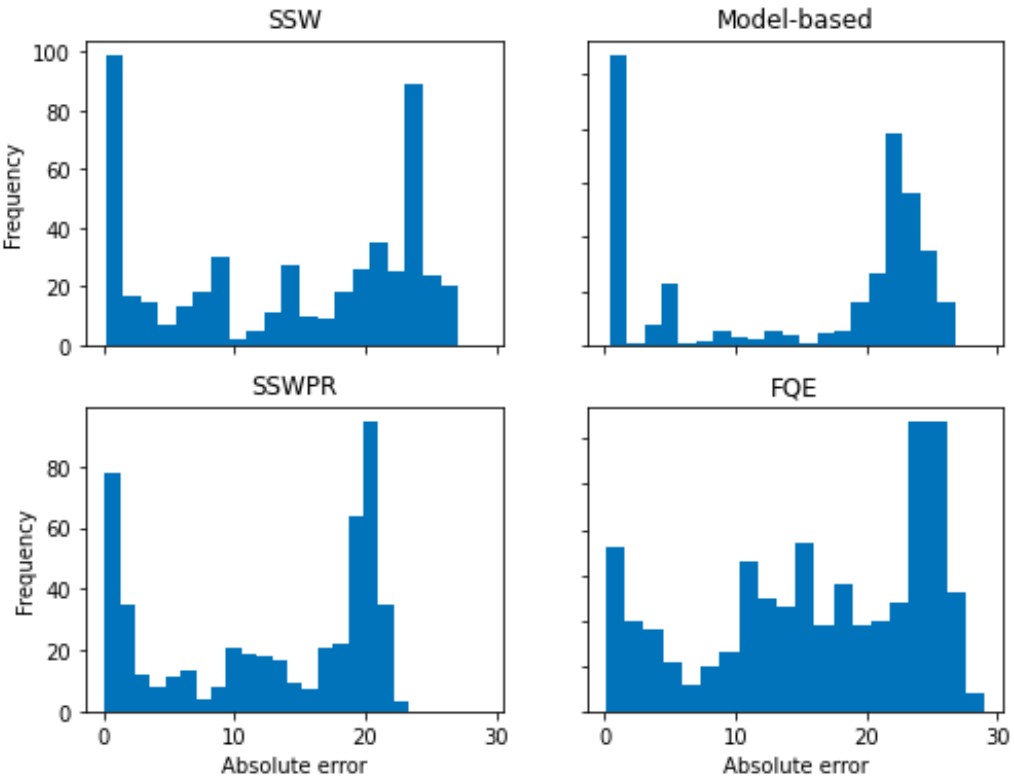

Figure 30: Histogram of absolute errors of SSW, SSWPR, FQE, and the model-based method for evaluating $\pi_{e_1}$ on the logged data deriving from $\pi_{b_2}$ on the mountain car environment

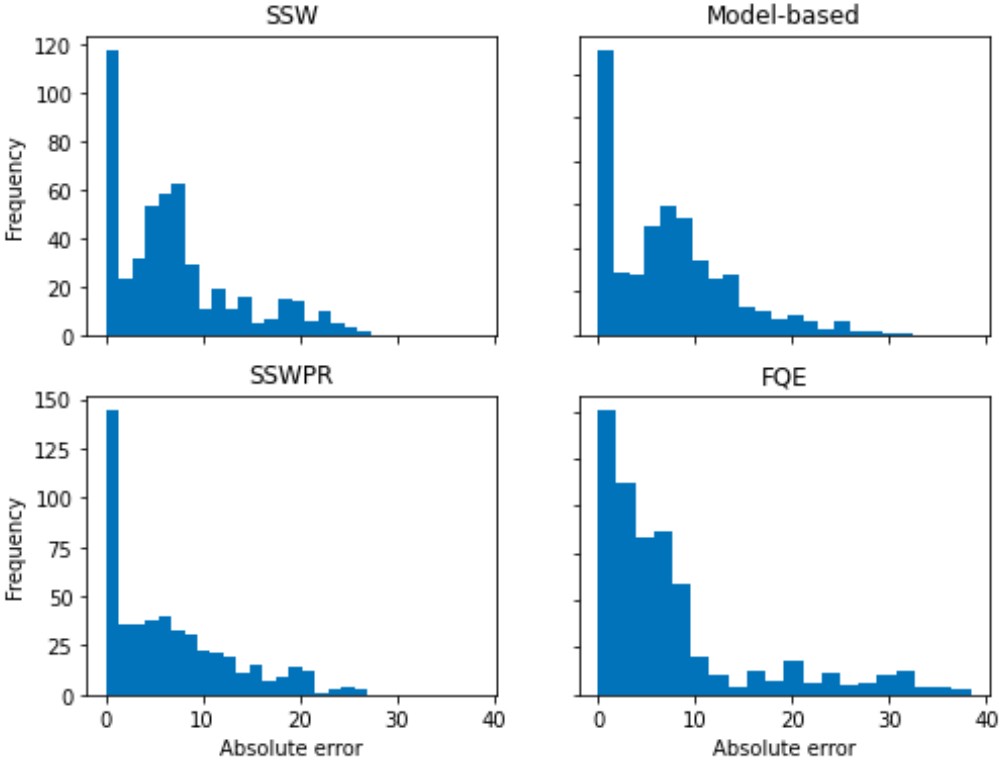

Figure 31: Histogram of absolute errors of SSW, SSWPR, FQE, and the model-based method for evaluating $\pi_{e_2}$ on the logged data deriving from $\pi_{b_2}$ on the mountain car environment

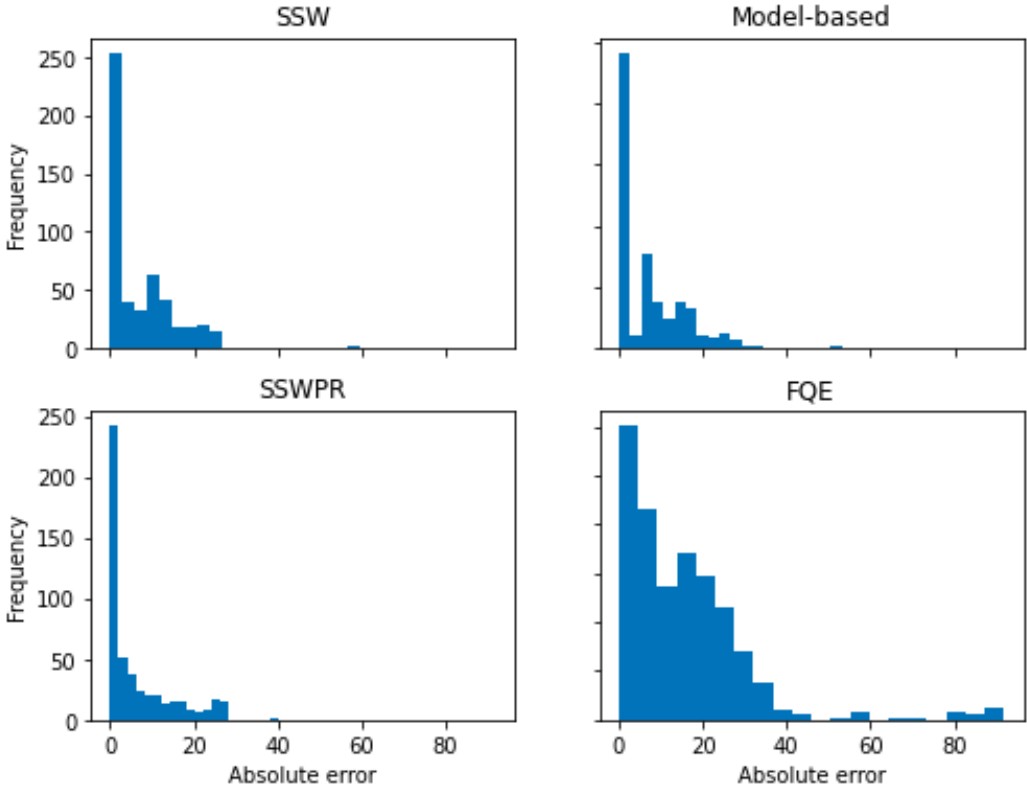

Figure 32: Histogram of absolute errors of SSW, SSWPR, FQE, and the model-based method for evaluating $\pi_{e_3}$ on the logged data deriving from $\pi_{b_2}$ on the mountain car environment

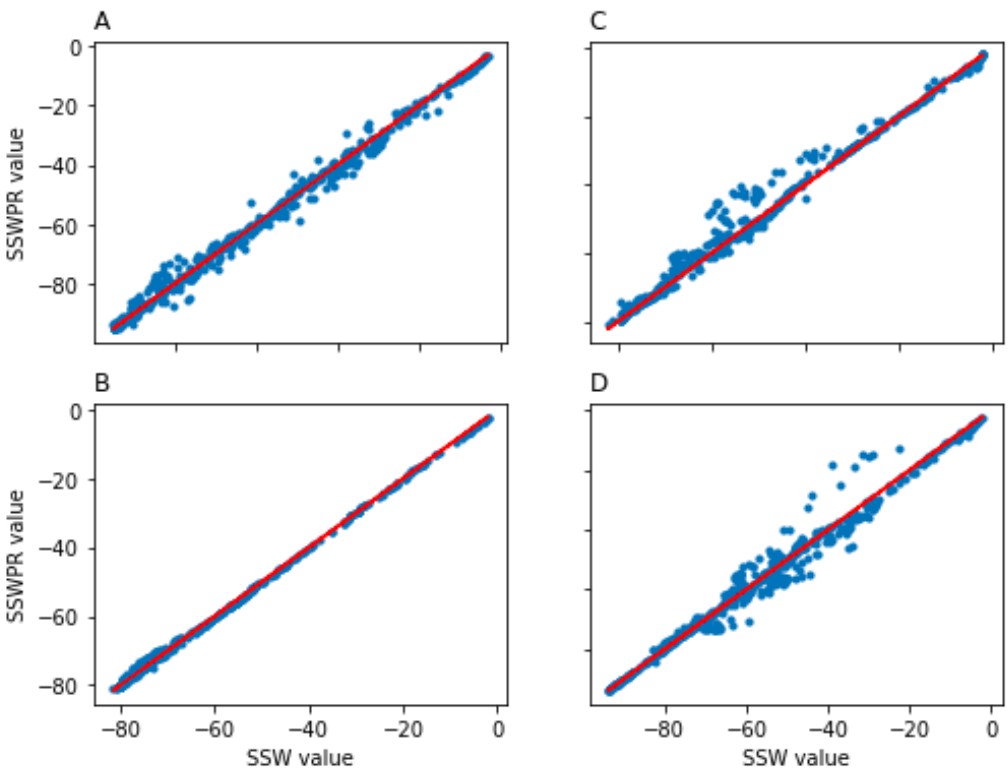

Figure 33: Scatterplots of the predictions of SSW vs SSWPR for evaluating A) $\pi_{b_1}$, B) $\pi_{e_1}$, C) $\pi_{e_2}$, D) $\pi_{e_3}$ on the logged data deriving from $\pi_{b_1}$ on the mountain car environment

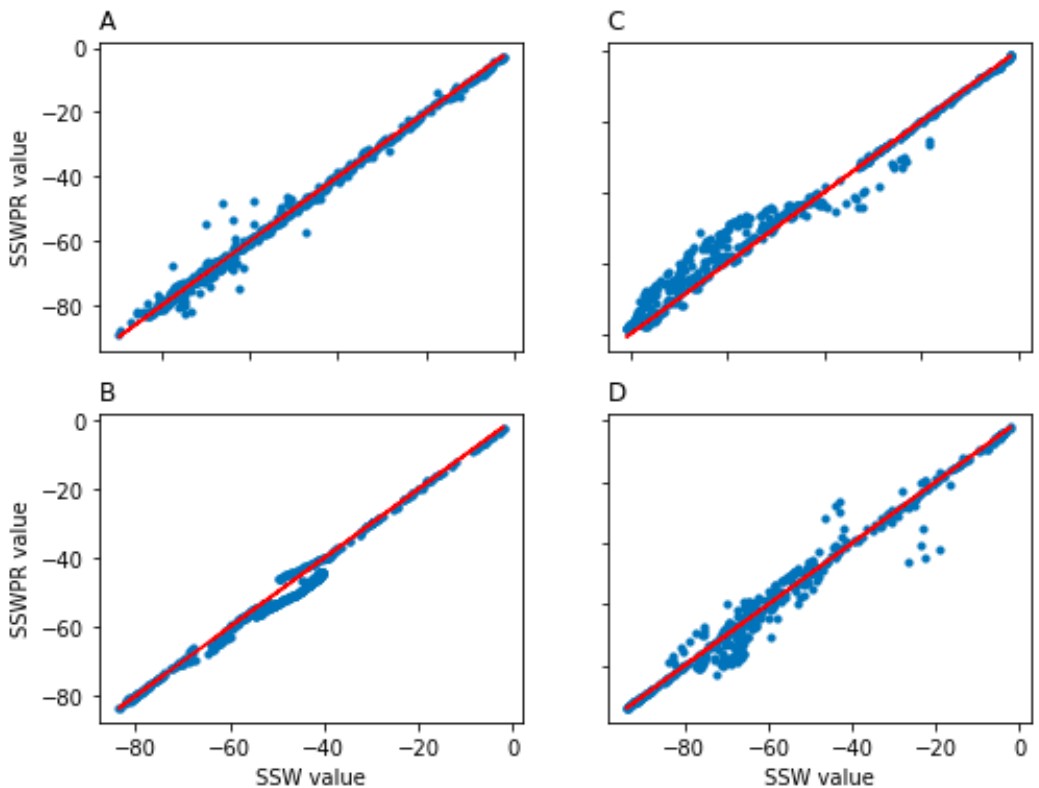

Figure 34: Scatterplots of the predictions of SSW vs SSWPR for evaluating A) $\pi_{b_2}$, B) $\pi_{e_1}$, C) $\pi_{e_2}$, D) $\pi_{e_3}$ on the logged data deriving from $\pi_{b_2}$ on the mountain car environment

