# OpenReview forum: "Offline evaluation in RL: soft stability weighting to combine fitted Q-learning and model-based methods"
_NeurIPS.cc/2022/Workshop/Offline_RL — Offline RL Workshop NeurIPS 2022_

### Official Review · Reviewer_zUMh · 2022-10-18
**Soft Stability Weighting for offline evaluation in RL**

**Rating:** 4
**Confidence:** 4

**Review:**

This paper proposed the soft stability weighting method for offline policy evaluation. It combined the FQE and model-based evaluation (MBE) methods. For each method, the author used ensembles and measure the stability using the standard deviation of the (s, a)-conditional predictions and the quantiles. The proposed method is evaluated using the 2D World and Mountain Car tasks.

Using the model ensemble to estimate uncertainty in the offline RL is not a novel idea [1, 2]. The relation of the proposed method with previous works needs to be discussed. The writing of this paper needs further improvement and the format used in this manuscript seems not correct. For example, in the section on Soft Stability Weighting, the symbols in the formula are not clearly explained. More importantly, the experimental task is the 2D World and Mountain Car which does not fully demonstrate the challenge of the offline policy evaluation problem. It's better to consider more complex robot control tasks.


[1] MOPO: Model-based Offline Policy Optimization

[2] Uncertainty-Based Offline Reinforcement Learning with Diversified Q-Ensemble

---

### Official Review · Reviewer_6Gcp · 2022-10-19

**Rating:** 4
**Confidence:** 4

**Review:**

This paper provides an adaptative weighting approach to combine FQE and model-based methods for off-policy evaluation. I have several concerns about this paper. 1) I cannot be convinced about the benefit of the proposed approach. Some theoretical analysis may be helpful. 2) the format of this paper looks quite informal. The is no section number, and the algorithm is just listed using numbered items. 3) The empirical results don't seem to be enough. The environment seems too simple. Given there is no theoretical evidence of the advantage of the proposed approach, I would suppose more comprehensive empirical results are required.